# Allosteric inhibition of HTRA1 activity by a conformational lock mechanism to treat age-related macular degeneration

Stefan Gerhardy [1], Mark Ultsch[2], Wanjian Tang [1], Evan Green[2], Jeffrey K. Holden[1], Wei Li[1], Alberto Estevez[2], Chris Arthur[2], Irene Tom[3], Alexis Rohou [2] & Daniel Kirchhofer [1] ✉

The trimeric serine protease HTRA1 is a genetic risk factor associated with geographic atrophy (GA), a currently untreatable form of age-related macular degeneration. Here, we describe the allosteric inhibition mechanism of HTRA1 by a clinical Fab fragment, currently being evaluated for GA treatment. Using cryo-EM, X-ray crystallography and biochemical assays we identify the exposed LoopA of HTRA1 as the sole Fab epitope, which is approximately 30 Å away from the active site. The cryo-EM structure of the HTRA1:Fab complex in combination with molecular dynamics simulations revealed that Fab binding to LoopA locks HTRA1 in a non-competent conformational state, incapable of supporting catalysis. Moreover, grafting the HTRA1-LoopA epitope onto HTRA2 and HTRA3 transferred the allosteric inhibition mechanism. This suggests a conserved conformational lock mechanism across the HTRA family and a critical role of LoopA for catalysis, which was supported by the reduced activity of HTRA1-3 upon LoopA deletion or perturbation. This study reveals the long-range inhibition mechanism of the clinical Fab and identifies an essential function of the exposed LoopA for activity of HTRA family proteases.

Age-related macular degeneration (AMD) is a chronic medical condition of the retina that leads to degeneration of the macula and causes progressive loss of central vision. From the early onset of AMD, characterized by accumulation of drusen droplets, AMD progresses to intermediate and late stages, leading to visual impairment and blindness. Depending on the histopathological features, AMD can be categorized into neovascular AMD and geographic atrophy (GA). While anti-VEGF agents are clinically used for the treatment of neovascular AMD, there are no treatment options available for GA. Pathologically, GA is characterized by a progressive loss of retinal photoreceptors, the retinal pigment epithelium (RPE), and choriocapillaris, resulting in atrophic lesions and central vision loss[1]. The onset and progression of AMD can be triggered by age, environmental factors, and genetics. Currently, more than 34 genetic markers have been associated with the risk of developing AMD[2,3]. Two major loci were commonly identified in

GWAS studies – the complement factor H locus (CFH) and the ARMS2/HTRA1 (Age-related maculopathy susceptibility 2/High temperature requirement A1) locus[2,4–6]. Genetic variation within the ARMS2/HTRA1 locus strongly correlates with an increased risk for both neovascular AMD and GA[2,7,8].

HTRA1 is a secreted protease composed of an N-terminal IGFBP-domain, a Kazal-like domain, a central trypsin-like serine protease domain and a C-terminal PDZ domain[9]. There are four closely related HTRA family members in humans, HTRA1-4, all of which share a conserved protease domain[10] and have been implicated in various pathologies[11–17]. HTRA1 itself cleaves a plethora of substrates and is linked to several pathogenic and developmental processes, including cerebral autosomal recessive arteriopathy with subcortical infarcts and leukoencephalopathy (CARASIL), AMD, Alzheimer's disease, osteoarthritis, neuronal development, and tumor progression[8,13,17–20].

[1]Department of Early Discovery Biochemistry, Genentech Inc., San Francisco, CA, USA. [2]Department of Structural Biology, Genentech Inc., San Francisco, CA, USA. [3]Department of OMNI Biomarker Development, Genentech Inc., San Francisco, CA, USA. ✉e-mail: dak@gene.com

HTRA1 protease activity is controlled through multiple mechanisms: First, HTRA1 requires trimerization of individual protomers to acquire full enzymatic activity. Genetic mutations within the monomer interface of HTRA1 that impede trimerization result in loss of activity associated with CARASIL[21,22]. Secondly, the N-terminal IGFBP and Kazal domains, as well as the C-terminal PDZ domain, are dispensable for catalysis[23,24]. Moreover, crystal structures of the HTRA1 protease domain demonstrate a substantial degree of plasticity within the active site region[23]. The protease domain adopts enzymatically non-competent and competent conformational states, leading to different models of HTRA1 regulation[23–25]. Our previously described structures of HTRA1 without bound substrate (apo-forms) suggest that HTRA1 activation follows the "conformational selection model", in which apo-HTRA1 can exist in various conformations: in the active, competent conformation and in multiple inactive, non-competent conformations[23]. In this model, active and inactive conformations exist in a dynamic equilibrium and substrates sample the catalytically active conformation, resulting in catalysis and product formation. Thus, shifting or disturbing this dynamic equilibrium among HTRA1 conformers could be a promising approach for therapeutic intervention.

The generation of specific and potent small molecule inhibitors of trypsin-like serine proteases, including HTRA1, remains a challenge in light of the structural conservation of active sites[26]. In contrast, antibodies are able to bind and neutralize serine proteases in a highly specific manner[27]. However, the trimeric nature of HTRA1, featuring a cradle-like arrangement of the three active sites in direct proximity, may prevent the simultaneous binding of antibodies or Fabs to all three actives sites. Therefore, effective inhibition of the HTRA1 enzymatic activity could be achieved either by dissociation of the trimer or by allosteric inhibition through binding to a more distant epitope. Dissociation of catalytically active oligomers has been reported for ß-tryptase, where an anti-tryptase antibody dissociates the ß-tryptase tetramer into inactive monomers[28,29]. For HTRA1, we have previously shown that an engineered Fab fragment (Fab94), binds to the surface exposed Loops B and C and inhibits HTRA1 enzymatic catalysis through an allosteric mechanism[30]. While the molecular mechanism of Fab94 remained unresolved, the study provided initial evidence that HTRA1 catalytic activity can indeed be targeted through a distant allosteric site. Following this rationale, we developed another series of antibodies with increased potency and inhibitory potential[31]. Hybridoma screening identified Fab15H6, which strongly inhibits the activity of human HTRA1 protease. The humanized and affinity-matured Fab15H6.v4.D226 was studied in pre-clinical models and in a Phase I clinical study[31–33]. It is currently being investigated as a novel treatment option for GA in a Phase II clinical trial (NCT03972709). However, the binding site and the mechanism by which the clinical Fab15H6.v4.D226 (FHTR2163, RG6147) inhibits HTRA1 enzyme function has remained obscure.

Here we describe the detailed inhibition mechanism of Fab15H6.v4 through a combination of cryogenic electron microscopy (cryo-EM), X-ray crystallography, molecular dynamics simulations, and biochemical assays. Our studies reveal that the Fab targets a small allosteric epitope on the distant LoopA of HTRA1. We found that this LoopA (37-Loop in the chymotrypsinogen numbering[34]) is essential for HTRA1 enzyme activity and that this LoopA function is shared by other human HTRA family members. Fab binding to LoopA locks several important active site loops in distorted and enzymatically non-competent conformations, completely disabling HTRA1 from recognizing or processing any of its substrates. This long-range allostery also controls other HTRA proteases, as transfer of HTRA1-LoopA to HTRA2 and HTRA3 enabled the Fab to inhibit these proteases. Therefore, the Fab15H6.v4 conformational lock mechanism establishes a paradigm for therapeutic approaches to target members of the mammalian HTRA family in a highly specific and effective manner.

## Results

### Inhibition of HTRA1 catalytic activity by Fab15H6.v4 is mediated through an allosteric mechanism

For all experiments described in this study, we used Fab15H6.v4, which differs from the clinical Fab15H6.v4.D221 by four additional residues at the C-terminus, which were removed in the clinical version to eliminate potential immunogenicity[31]. We first used the established casein-BODIPY protease assay to examine the enzymatic inhibition by Fab15H6.v4 of full-length HTRA1 (HTRA1$^{FL}$, 51 kDa) and the protease domain alone (HTRA1$^{PD}$, 24 kDa) (the catalytically inactive S328A mutant HTRA1$^{PD/SA}$ served as negative control). The results showed that Fab15H6.v4 inhibited both HTRA1$^{FL}$ and HTRA1$^{PD}$ by >90%, while enzyme activity was unaffected in the presence of the control Fab33 (anti-PCSK9,[35]; Fig. 1a). We also investigated the inhibitory activity of Fab15H6.v4 towards HTRA1-mediated processing of several known macromolecular substrates, including the previously described biomarker DKK3 (Dickkopf-related protein 3,[31]) as well as BIGLYCAN and DECORIN[23,36]. We have used a low substrate: enzyme ratio (4:1) to obtain complete cleavage of these substrates and to assess the inhibition by Fab15H6.v4 under stringent conditions (see Supplementary Fig. 1a–d for HTRA1 time- and concentration-dependent cleavage of the three substrates). Like in the generic substrate assay, Fab15H6.v4 inhibited cleavage of these macromolecular substrates by HTRA1$^{FL}$ and HTRA1$^{PD}$ (Fig. 1b). Lastly, we used a small fluorescent activity-based probe (TAMRA-Leu-Val-phosphonate; TAMRA-ABP[31]), that covalently binds to the catalytic serine S328, to test accessibility to the active site of HTRA1. Fab15H6.v4 completely inhibited labeling of both HTRA1$^{FL}$ and HTRA1$^{PD}$ (Fig. 1c), indicating that the Fab-bound HTRA1 is unable to bind a small substrate mimetic. These findings suggested that the inhibition mechanism of the Fab is not substrate-specific but rather affects fundamental aspects of enzyme function.

Since only a stable HTRA1 homotrimer can support full catalytic activity[21,22], we considered the possibility that Fab15H6.v4 might dissociate the HTRA1 trimer in analogy to an anti-tryptase antibody, which was shown to inhibit catalysis by dissociating the tryptase tetramer into inactive monomeric subunits[29]. By use of size-exclusion chromatography (SEC), we found that Fab15H6.v4 did not dissociate trimers of HTRA1$^{PD}$, or of catalytically inactive S328A mutants HTRA1$^{FL/SA}$ or HTRA1$^{PD/SA}$ (Supplementary Fig. 3a, fig. 1d). Instead, the molecular weights of the formed complexes increased, indicating a 1:3 stoichiometry (HTRA1 trimer:Fab15H6.v4), where one Fab bound to each protomer of the HTRA1 trimer (Fig. 1d). In addition, titration of HTRA1$^{PD}$ with Fab15H6.v4 showed a progressive inhibition of HTRA1$^{PD}$ enzyme activity at sub-stoichiometric concentration, reaching complete inhibition at a 1:3 (HTRA1 trimer:Fab15H6.v4) molar ratio (Supplementary Fig. 1e), consistent with the stoichiometry determined by size exclusion chromatography (Fig. 1d).

Furthermore, we investigated whether Fab15H6.v4 might bind to the active site itself. When the active site of HTRA1$^{PD}$ was blocked with a heptameric activity-based probe (biotin-DPMFKLV-phosphonate derived from[24], 7-mer ABP), Fab15H6.v4 still bound with sub-nanomolar affinity (HTRA1$^{PD}$ + 7-mer ABP: $K_D$ = 0.21 nM, compared to the uninhibited HTRA1$^{PD}$: $K_D$ = 0.12 nM, Fig. 1e, Supplementary Table 1), indicating that the Fab does not bind to or near the HTRA1 active site itself, but to an allosteric site. The relatively small decrease in the affinity of Fab15H6.v4 to the active-site blocked HTRA1$^{PD}$ is consistent with the reported 1.5 to 3-fold affinity reductions to active site-blocked urokinase-type plasminogen activator (uPA) by allosteric antibodies[37,38]. Moreover, we tested a small single chain Fv (scFv) version of Fab15H6.v4, which had reduced binding affinity (HTRA1$^{PD}$ $K_D$ = 9.4 nM, Supplementary Fig. 1f, Supplementary Table 1), but efficiently inhibited HTRA1$^{FL}$ and HTRA1$^{PD}$ (Supplementary Fig. 1g) and blocked binding of the fluorescent TAMRA-ABP (Supplementary Fig. 1h). This indicated that inhibition by Fab15H6.v4 is not due to steric hindrance of substrate access to the active site. Together, these

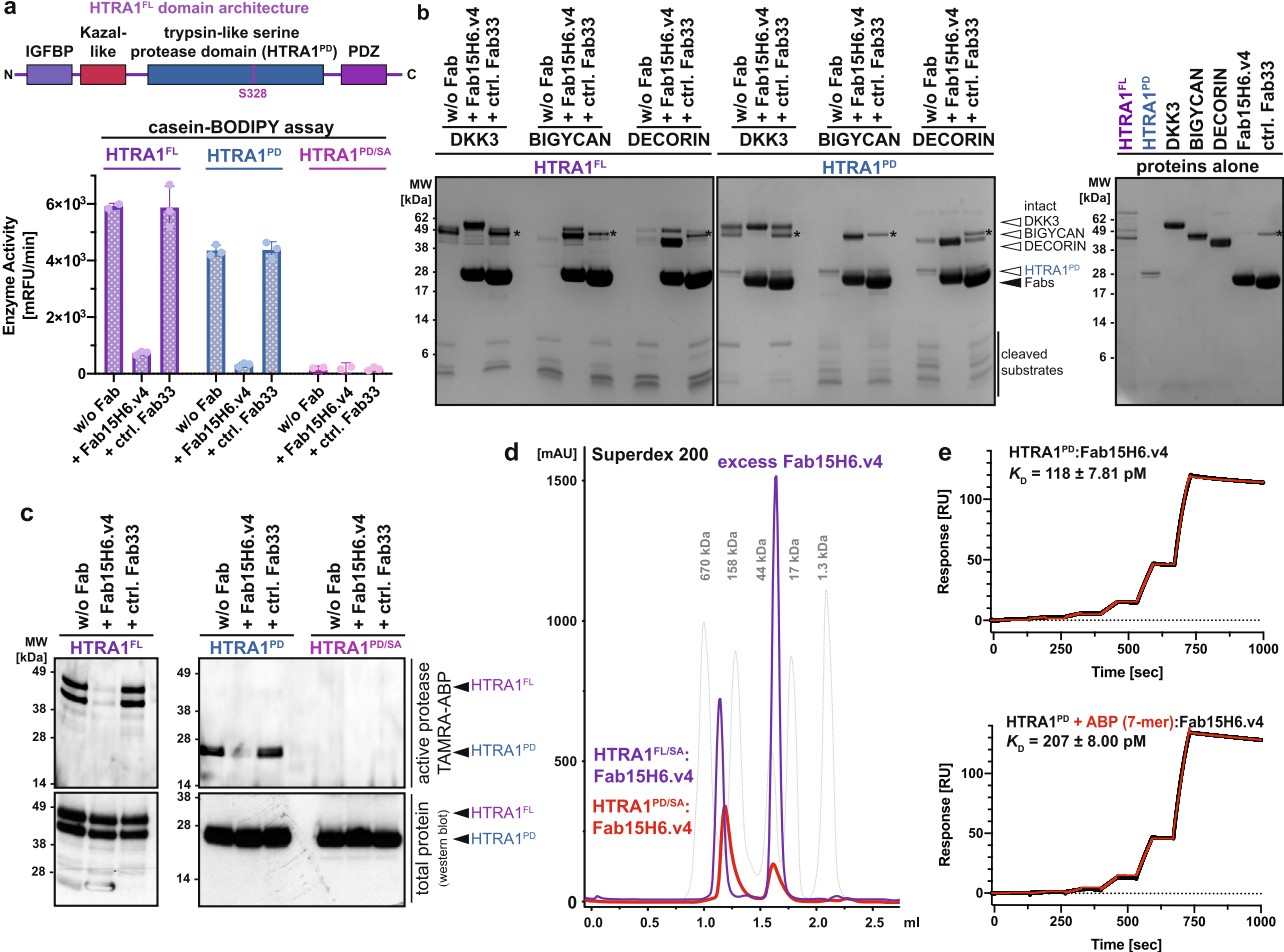

**Fig. 1 | Fab15H6.v4 inhibits the enzymatic activity of HTRA1^FL and HTRA1^PD through an allosteric mechanism. a** HTRA1^FL domain architecture. Enzyme activity of HTRA1^FL, HTRA1^PD, and catalytically inactive HTRA1^PD/SA in the presence of Fab15H6.v4. Control Fab33 (anti-PCSK9) does not affect enzymatic activity. **b** Cleavage of macromolecular substrates Dickkopf-related protein 3 (DKK3), BIGLYCAN, and DECORIN by HTRA1^FL (51 kDa) and HTRA1^PD (24 kDa) in the presence of Fab15H6.v4. Right panel: cleavage assay control with individual proteins. Asterisks (*) indicate the contaminant of the control Fab33 preparation. **c** Labeling of HTRA1^FL, HTRA1^PD, and HTRA1^PD/SA active site using a small fluorescent activity-based probe (TAMRA-ABP) in the presence of Fab15H6.v4. HTRA1^FL undergoes self-

cleavage and therefore appears as two bands in the TAMRA-ABP assay. **d** Size-exclusion profiles of HTRA1^PD/SA:Fab15H6.v4 (red, calc. mass = 228 kDa) and full length HTRA1^FL/SA:Fab15H6.v4 (blue, calc. mass = 311 kDa) complexes. Protein standards for estimated size comparison are in grey. **e** Kinetics of Fab15H6.v4 binding to HTRA1^PD and to HTRA1^PD pre-incubated with 7-mer ABP determined by Surface Plasmon Resonance (SPR). Bar graphs in (**a**) and kinetic data in (**e**) are presented as the mean ± S.D. of three independent experiments. Images in (**b, c**), as well as chromatogram in (**d**) are representative of two independent experiments. Source data are provided as a Source Data file.

findings suggested that Fab15H6.v4 binds and inhibits HTRA1 in its trimeric form at an allosteric site and interferes with fundamental aspects of the catalytic machinery.

## Cryo-EM structures reveal an allosteric inhibition mechanism originating from the exposed LoopA of HTRA1

In order to gain insight into the Fab inhibition mechanism, we determined structures of the HTRA1:Fab15H6.v4 complex using single particle cryo-EM. First, we investigated the complex of the Fab bound to the catalytically inactive HTRA1^PD/SA mutant (Fig. 2a, Supplementary Fig. 2 and Supplementary Table 2). To confirm that the observed Fab-induced structural changes in the active site region of HTRA1^PD/SA were not caused by the mutated catalytic serine residue, we also solved a structure of Fab-bound wildtype HTRA1^PD (Fig. 2b, Supplementary Fig. 3 and Supplementary Table 2). We obtained structures of the catalytically inactive HTRA1^PD/SA and wildtype HTRA1^PD at 3.3 Å resolution. In addition, we determined a crystal structure of apo Fab15H6.v4 to a resolution of 1.8 Å (Supplementary Table 3), which we fitted into the cryo-EM maps. Atomic structures built into the cryo-EM maps of the wildtype and catalytically inactive HTRA1^PD:Fab complexes were

similar in their loop distortions (Supplementary Fig. 4a), indicating that the alanine mutation of the catalytic serine S328 had no adverse structural influence.

In both structures, one Fab15H6.v4 binds to each of the three protomers of the trimeric HTRA1, consistent with the 1:3 stoichiometry (HTRA1 trimer:Fab15H6.v4) determined by SEC and HTRA1^PD titration experiments (Figs. 1d, S1e, S3a). The binding epitope is located to the exposed LoopA (37-Loop in the chymotrypsinogen numbering system[34]) (Fig. 2c, d), which is more than 30 Å away from the active site, and is contacted exclusively by the heavy chain CDR1, CDR2 and CDR3, without any contributions from the light chain of the Fab (Fig. 2c). The LoopA epitope within HTRA1^PD is positively charged and interacts with the negatively charged paratope of Fab15H6.v4 (Supplementary Fig. 4b, c). In comparison to the crystal structures of unbound apo-HTRA1, the LoopA in both HTRA1:Fab15H6.v4 complexes is twisted outwards by almost 20° and the β-strand stem partially unfolds at V199-V201 into a loop structure (Supplementary Fig. 4d).

Even though the Fab15H6.v4 binding site is more than 30 Å away from active site it profoundly influences the arrangement of the active center loops (LoopD, Loop1 and Loop2; 140-Loop, 180-Loop and 220-

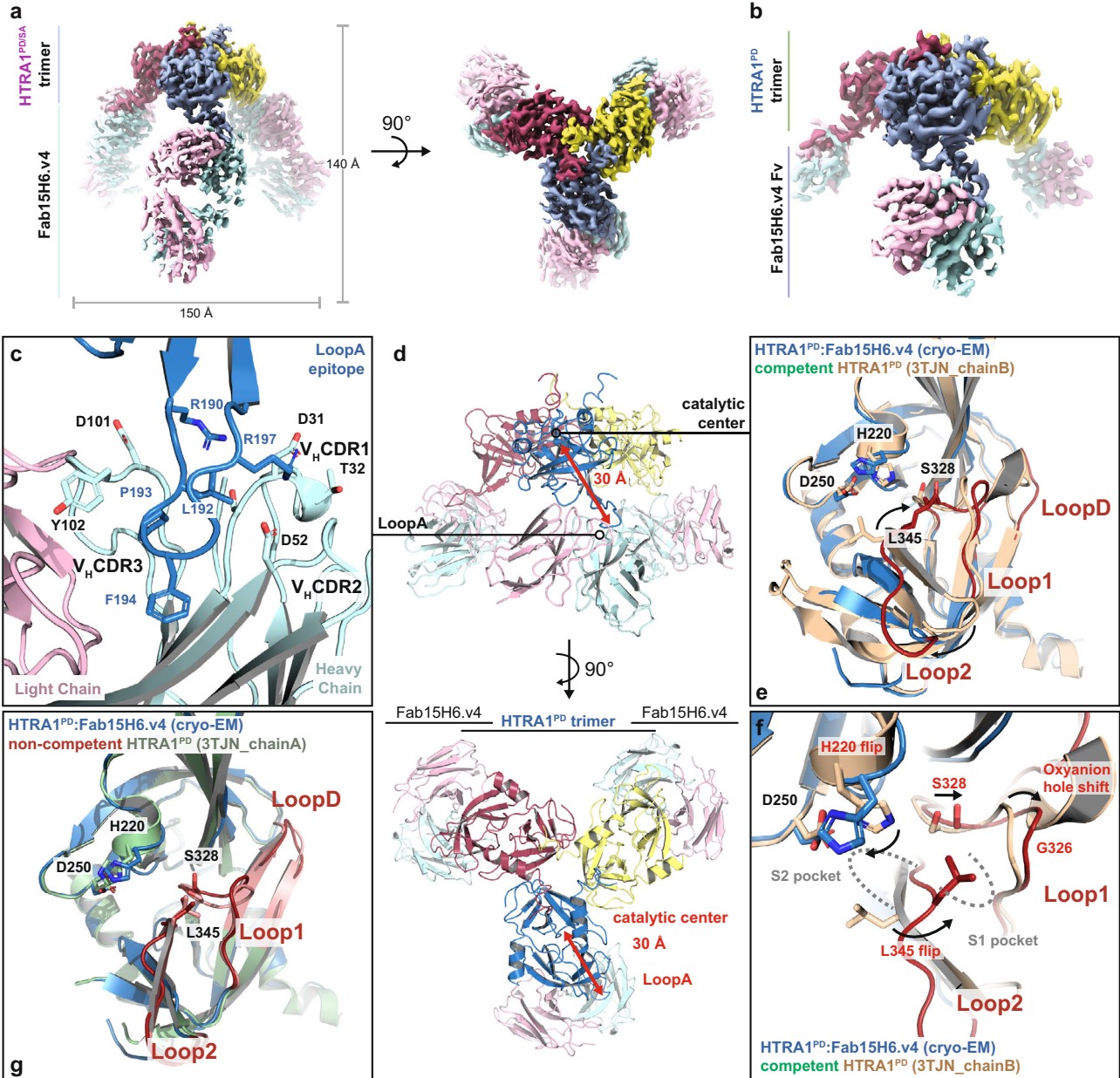

**Fig. 2 | Fab15H6.v4 binds to the distant LoopA of the HTRA1 trimer and stabilizes a non-competent conformation. a** Cryo-EM map of HTRA1[PD/SA] trimer with three Fabs (Fab15H6.v4) bound to the exposed LoopA at a resolution of 3.3 Å. The monomers of the HTRA1[PD/SA] trimer are in blue, yellow and red and Fab light and heavy chains are in pink and cyan, respectively. **b** Cryo-EM map of wildtype HTRA1[PD]:Fab15H6.v4 complex at 3.3 Å resolution, in which the flexible constant regions ($C_H$I/$C_L$) of the Fab15H6.v4 were masked out during refinement. **c** Detailed view on the LoopA epitope and Fab15H6.v4 interaction. Critical residues are represented as sticks. **d** Ribbon representation of wildtype HTRA1[PD] bound by Fab15H6.v4 in side view (top) and top view (bottom). The LoopA epitope is more than 30 Å away from the catalytic center. **e** Comparison of the Fab-bound wildtype HTRA1[PD] (blue) with the competent apo-HTRA1 in the active conformation (wheat, pdb 3TJN_chainB) showing severe loop distortions in the catalytic center (loops in red). **f** Detailed view of the distorted active center loops (red) and catalytic residues of Fab-bound wildtype HTRA1[PD] (blue) in comparison to the competent apo-HTRA1 conformation (wheat, pdb 3TJN_chainB); the catalytic serine S328 and H220 are out of position, the oxyanion hole (indicated by residue G326) is not formed and the L345 occludes the S1 pocket. **g** Comparison of Fab-bound HTRA1[PD] (blue) and apo-HTRA1[PD] (green, pdb 3TJN_chainA) with LoopD, Loop1, and Loop2 in similar, non-competent conformations (red, salmon).

Loop in the chymotrypsinogen numbering system). In both structures, the wildtype HTRA1[PD] and the mutant HTRA1[PD/SA] (Figs. 2e and S5a), these loops adopt conformations which are incompatible with catalysis and strongly differ from the arrangement seen in the catalytically competent active site of HTRA1 (wheat in Figs. 2e and S5a,[23,24]). Both Loop1 and Loop2 in our Fab-bound structures are severely distorted, resulting in a non-functional catalytic triad, an obstructed S1 specificity pocket and a non-functional oxyanion hole (Fig. 2f, S5b). The distorted Loop1 is shifted towards LoopD, moving the catalytic serine S328 away from its canonical position (Figs. 2f, S5b). In addition, the catalytic

H220 side chain is flipped away from the catalytic serine and is therefore unavailable to function as hydrogen acceptor during catalysis (Figs. 2f, S5b). Moreover, the important oxyanion hole residue G326 has moved more than 4 Å away from its canonical position and will not be able to stabilize the tetrahedral intermediate (Fig. 2f, S5b). Lastly, the failure to orient Loop2 correctly has a major impact on substrate binding: the distorted Loop2 cannot engage in the canonical ß-strand interactions with the substrate and the flipped side chain residue L345 occludes the S1 pocket. Together these altered conformations prevent access of the substrate P1 residue to the active site

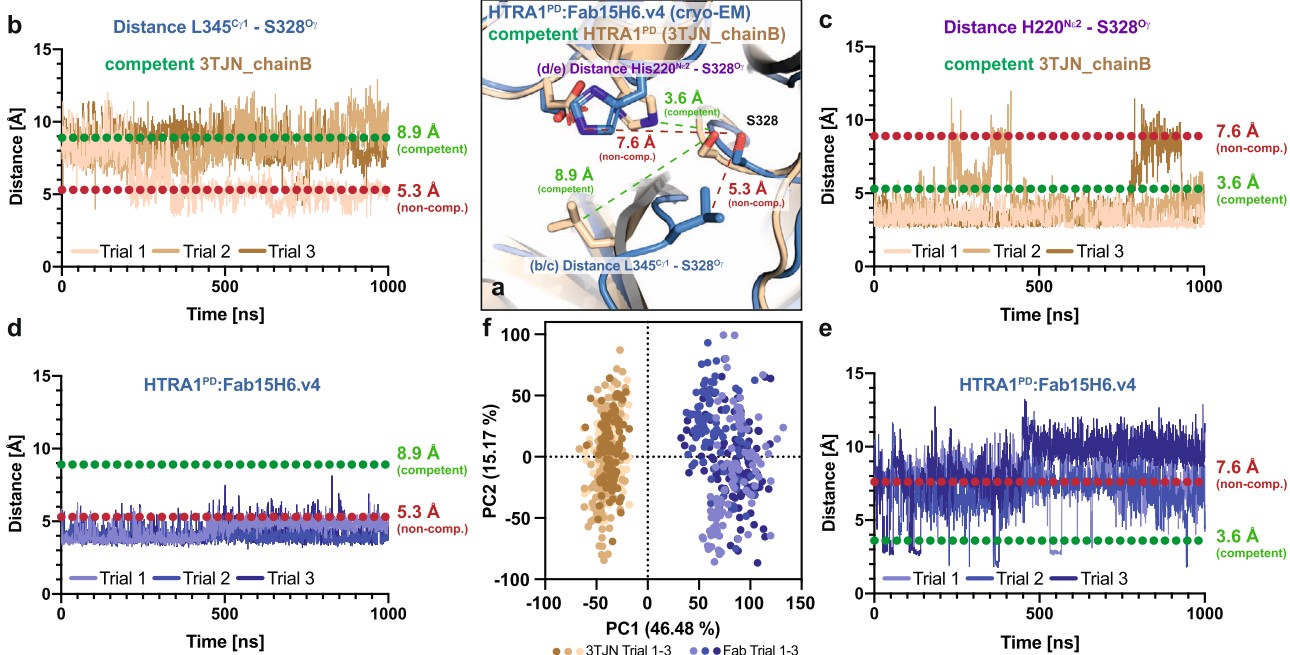

**Fig. 3 | Molecular dynamics (MD) simulations show that binding of Fab15H6.v4 to LoopA locks HTRA1 in a non-competent state. a** Detailed conformational differences within the catalytic center of competent (brown, pdb 3TJN_chainB) and Fab-bound HTRA1$^{PD}$ (blue). The distance (dashed line) between L345 and S328 decreases from competent (green) to the non-competent conformation (red) where it occludes the S1 pocket. The distance (dashed line) between H220 and S328 increases when transitioning from the active (green) to the inactive (red) conformation, impeding the hydrogen transfer during catalysis. **b, c** Measured distance between L345$^{C\gamma1}$ and S328$^{O\gamma}$ (**b**) and between H220$^{N\epsilon2}$ and S328$^{O\gamma}$ (**c**) during the MD simulations of competent HTRA1$^{PD}$ (wheat) with the calculated distances of competent and non-competent conformation indicated by green and red dotted lines, respectively. **d, e** Measured distance between L345$^{C\gamma1}$ and S328$^{O\gamma}$ (**d**) and between H220$^{N\epsilon2}$ and S328$^{O\gamma}$ (**e**) during the MD simulations of Fab-bound HTRA1$^{PD}$ (blue). Expected distances for competent and non-competent conformations are indicated by green and red dotted lines, respectively. **f** Principal component analysis of MD simulations of 3TJN_chainB and the Fab-bound HTRA1$^{PD}$, showing cross plots of the first two individual principal components.

(Fig. 2f, S5b). In the active conformation (Fig. 2f, in wheat), the side chain of L345 is rotated outwards, which opens the S1 pocket and enables substrate binding[23,24,39].

Each one of these conformational changes seen in the HTRA1:Fab complexes would be detrimental to enzymatic catalysis and collectively these structural changes render HTRA1 non-competent and unable to cleave any substrate or to bind small activity-based probes.

## Binding of Fab15H6.v4 disables transition of HTRA1 to the enzymatically competent state by a conformational lock mechanism

The observed non-competent arrangement of Loops 1, 2 and D in our cryo-EM structures is almost identical to those of the published crystal structures of apo-HTRA1$^{PD/SA}$ (e.g. pdb 3NUM_chainA and 3TJO_chainX[23]) and similar to the non-competent structures of wild-type apo-HTRA1$^{PD}$ (pdb 3NWU and 3TJN_chainA, Figs. 2g, S5c). However, in the context of unbound HTRA1 trimers used in crystallography studies, these non-competent conformers can transition to the enzymatically competent state and engage in catalysis as observed for wildtype apo-HTRA1[23]. Therefore, we wondered whether similar conformational changes could occur in the Fab-bound state.

Careful examination of our cryo-EM map revealed that features of the catalytic site were resolved as well as, if not better, than most other regions of the maps. More quantitatively, local resolution estimates were the highest within the catalytic centers (Supplementary Figs. 2e, f and S3e, f), indicating the absence of detectable conformational heterogeneity within these regions. The increased resolution within the catalytic center allowed confident assignments of sidechain orientations, including the catalytic residues H220 and D250 (Supplementary Fig. 3h). Similarly, a clear density of the flipped L345 side chain occluding the S1 pocket was observed and allowed confident sidechain

placement. Consistent with the lack of detectable conformational flexibility in this region, atomic B factors were the lowest within the catalytic centers (Supplementary Fig. 3i). Finally, we also re-processed our datasets in the absence of any imposed point-group symmetry and failed to resolve any deviations from three-fold symmetry within the catalytic center (Supplementary Figs. 2e, f and 3e, f).

We concluded that the non-competent conformation observed in all our cryo-EM maps is the dominant conformational state of HTRA1 when Fab15H6.v4 is bound and that the transition to the enzymatically competent state does not occur to any detectable degree in the Fab-bound condition.

We further tested this model by employing molecular dynamics (MD) simulations of previously published structures[23,24], the apo-HTRA1 (pdb 3TJN_chainB) and HTRA1 with a covalently bound substrate mimic (pdb 3NZI_chainA), which was removed for our simulations. These two structures, both representing the competent state, were compared with our Fab-bound wildtype HTRA1$^{PD}$ structure. We focused our analysis on two distinct conformational changes that render the active center non-competent (Fig. 3a): The occlusion of the S1 pocket by L345 of Loop2 and the flip of the catalytic triad H220 side chain. In comparison to the competent conformation these perturbations significantly change the distances between the catalytic serine and the side chains of H220 and L345. We continuously monitored the distances between these reference points throughout the MD simulations.

In the two apo-HTRA1 structures (pdb 3TJN_chainB and 3NZI) both reference points were mostly found in the competent conformation with an open S1 pocket and correctly positioned H220 (Figs. 3b, c and S6a, b). In one of three MD simulation runs of the apo-HTRA1 (Fig. 3b, pdb 3TJN_chainB, Trial 1), we noticed that the L345 side chain had flipped, in agreement with the observed flexibility of the Loop2 in the

reported crystal structures[23]. In contrast, the Fab-bound HTRA1[PD] was unable to sample the active conformation and remained in the non-competent conformation characterized by a flipped L345 side chain occluding S1 access (Fig. 3d, red). Similarly, H220 was found predominantly in the non-competent conformation (Fig. 3e) and consequently reduces the nucleophilicity of the catalytic S328. The observed differences in the active site conformational dynamics would explain why the Fab15H6.v4 completely prevented adduct formation of a small activity-based probe (TAMRA-ABP, Fig. 1c). Deconvolution of each MD trajectory by principal component analysis also revealed the conformational sampling between each trajectory to be quite different between the Fab bound structure and 3TJN (Fig. 3f).

Taken together, our cryo-EM structures and MD simulations demonstrate that binding of Fab15H6.v4 locks the HTRA1 active center loops in an enzymatically non-competent state, thereby disabling the transition of HTRA1 to an active state, resulting in complete inhibition of catalysis.

### The LoopA provides exquisite binding specificity of Fab15H6.v4 for HTRA1

We further validated the importance of the HTRA1-LoopA for Fab15H6.v4 binding by deleting the entire LoopA and replacing it with a beta hairpin motif "NG" (HTRA1[PD]-ΔLoopA)[40]. This HTRA1[PD]-ΔLoopA construct did not show any detectable binding to Fab15H6.v4 (Fig. 4a), confirming that LoopA comprises the entire binding epitope. Next, we determined the co-crystal structure of the LoopA peptide bound to Fab15H6.v4 at a higher resolution (2.1 Å) than our cryo-EM structures, which allowed us to identify the precise molecular interactions between LoopA and the Fab15H6.v4. The LoopA peptide aligned well with LoopA from our cryo-EM structure and bound to the exact same paratope region and with similar orientations of side chains contacting CDR1, CDR2 and CDR3 of the Fab (Fig. 4b, c). Compared to our solved apo-Fab structure, LoopA binding only slightly reoriented CDR2, while the other CDRs and the overall architecture remained unchanged (Supplementary Fig. 7a, b). Our structure allowed confident identification of the key residues (R190, L192, P193, F194 and R197) mediating the interaction between the Fab and LoopA of HTRA1 (Fig. 4c, d). Alanine mutations of each of these residues in the HTRA1[PD] protein as well as in synthetic LoopA peptides confirmed their role in binding to Fab15H6.v4 by SPR (Supplementary Tables 4 and 5), except for the F194A mutant, which had reduced binding in the peptide format only. Conversely, alanine mutations of key Fab15H6.v4 paratope residues within the three CDR loops (Fig. 4d, blue) resulted in reduced HTRA1[PD] binding, but complete binding loss required mutation of all eight residues (Supplementary Table 6). The two mutants D52A and Y100A had the greatest affinity losses (2000-fold and 200-fold, respectively), likely due to the importance of D52 and Y100 in interacting with the key LoopA residues R197 and R190, respectively.

The finding that LoopA comprised the entire epitope allowed us to determine the specificity of Fab15H6.v4 by measuring its binding to synthetic peptides encompassing the LoopA sequences of the HTRA family members HTRA1-4. The HTRA1-LoopA peptide bound with high affinity to Fab15H6.v4 ($K_D = 0.77$ nM), whereas the LoopA peptides of HTRA2-4 showed no detectable binding (Fig. 4e, Supplementary Table 5), in agreement with a previous study using HTRA1-4 proteins[31]. In an orthogonal experiment, we demonstrated three chimeric HTRA1 proteins in which the LoopA was replaced with that of HTRA2, HTRA3, and HTRA4 (HTRA1-LoopA[HTRA2/3/4]) had no detectable binding to Fab15H6.v4 (Fig. 4f, g). Sequence alignment of the human HTRA family (Fig. 4h) and a conservation map (Supplementary Fig. 7c, d) showed that the LoopA is the least conserved region of the otherwise highly conserved protease domains. LoopA of HTRA1 differs from LoopA of HTRA2, HTRA3 and HTRA4 by 5−7 amino acids (Fig. 4h). Additionally,

superpositions of the LoopA from the crystal structures of HTRA2 and HTRA3 with that of Fab15H6.v4-bound LoopA of HTRA1 provided a structural explanation for the Fab specificity (Supplementary Fig. 7e). Changes in key interaction residues and interference with neighboring loops in the other family members provide exclusive specificity of Fab15H6.v4 for HTRA1.

The HTRA1 protease domain including the LoopA epitope is highly conserved across mammalian species (Supplementary Fig. 8a,[10]). Thus, our structural model predicts that Fab15H6.v4 could also be used in different disease model organisms. Indeed, Fab15H6.v4 inhibits murine HTRA1 (Supplementary Fig. 8b) and blocks the labeling of its active site by TAMRA-ABP (Supplementary Fig. 8c) similar to human HTRA1 (Fig. 1).

These results demonstrate that LoopA embodies the complete Fab epitope, and a combination of sequence-specific interactions and unique structural features of LoopA confer binding specificity for mammalian HTRA1.

### LoopA is indispensable for the catalytic activity of HTRA family proteases

The inactivation of HTRA1 enzyme function by Fab binding raised the possibility that LoopA is an important contributor to enzymatic activity. Consistent with this hypothesis, the HTRA1[PD]-ΔLoopA mutant had greatly reduced enzymatic activity (<30% of wildtype activity) in the casein-BODIPY assay (Fig. 5a). To ascertain that the impaired activity was not due to trimer destabilization, we performed a size-exclusion chromatography experiment, which showed that HTRA1[PD]-ΔLoopA formed stable trimers (Fig. 5b). We constructed the corresponding deletion mutants of HTRA2 and HTRA3 (HTRA2[PD/PDZ]-ΔLoopA and HTRA3[PD/PDZ]-ΔLoopA), to find out whether LoopA is also essential for other HTRA proteases. Both mutants were still able to form stable trimers (Fig. 5b and Supplementary Fig. 8d) but, like HTRA1-ΔLoopA, showed a strongly reduced enzymatic activity (<25% of wildtype activity) in the casein-BODIPY assay (Fig. 5a). As expected, the activity of HTRA2, HTRA3, their LoopA deletion mutants as well as HTRA1-ΔLoopA was unaltered in the presence of the Fab15H6.v4, which is unable to recognize the Loop deletion mutant of HTRA1 (Fig. 4a) or the other HTRA family members (Fig. 4e).

Next, we used the chimeric proteins HTRA1-LoopA[HTRA2/3/4] to understand the importance of the HTRA1-LoopA sequence in promoting enzyme activity. We found that all LoopA chimeras (HTRA1-LoopA[HTRA2/3/4]) showed a very strong reduction in enzyme activity, particularly HTRA1-LoopA[HTRA3] and HTRA1-LoopA[HTRA4] (Fig. 5c), both of which have 7 amino acid changes compared to wildtype HTRA1-LoopA, whereas HTRA1-LoopA[HTRA2] has only 5 changes. As expected, the Fab15H6.v4 did not further reduce the activity (Fig. 5c), because it no longer bound to the chimeric proteins (Fig. 4f, g). Lastly, in agreement with the casein-BODIPY assay results, the HTRA1-ΔLoopA and HTRA1-LoopA[HTRA2] mutants displayed strongly reduced activities in the in-vitro DKK3 cleavage assay (Fig. 5d, e), whereas HTRA1-LoopA[HTRA3] and HTRA1-LoopA[HTRA4] were unable to cleave DKK3 (Supplementary Fig. 8e).

Furthermore, individual residues within LoopA were mutated to alanine to identify the critical LoopA residues required for HTRA1 activity. While the effects on enzymatic activity were less dramatic compared to the LoopA chimeras results, all mutants exhibited reduced activities down to about 60% and 50% of wildtype activity for the F194A and R197A mutant, respectively (Fig. 5f). We suspected that the function of LoopA is associated with its conformational dynamics that is linked to the active site. Therefore, to disrupt this connectivity we inserted a flexible GSG or GSGSG linker at the juncture between LoopA and the β-strand stem (Fig. 5f). Both mutants displayed strongly reduced enzymatic activities (<20% of wildtype, Fig. 5f). Lastly, to verify that LoopA is also requisite for the

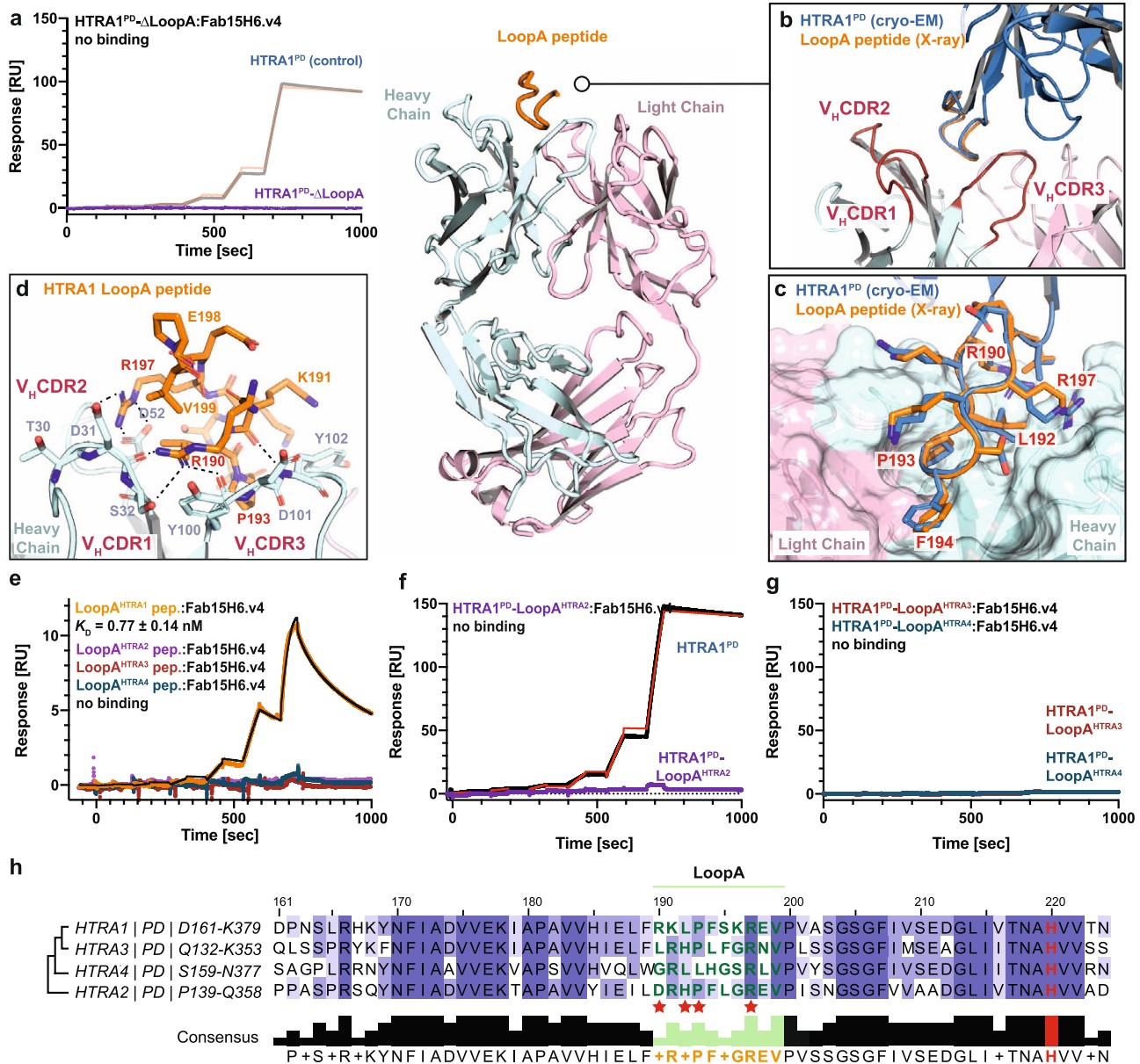

**Fig. 4 | Fab15H6.v4 binds to HTRA1 LoopA epitope with high specificity. a** SPR binding kinetics show no interaction between Fab15H6.v4 and HTRA1-ΔLoopA mutant. Overview of the co-crystal structure of Fab15H6.v4 with LoopA peptide (center). **b** Detail of co-crystal structure showing an excellent alignment of the LoopA peptide (orange) with the LoopA of the cryo-EM structural model (blue). In the co-crystal structure the LoopA is exclusively contacted by the Fab heavy chain (CDR1-3, red). **c** Side chain comparison between LoopA peptide of the co-crystal structure (orange) and LoopA of the cryo-EM structural model (blue) with key residues labeled in red. **d** Key residues between LoopA and Fab15H6.v4 based on the co-crystal structure include R190, L192, P193 and R197 (labeled red). **e** SPR binding kinetics of immobilized Fab15H6.v4 to LoopA peptides derived from HTRA family members. **f, g** SPR binding kinetics experiments show no interaction between Fab15H6.v4 and the chimeric proteins HTRA1-LoopA^HTRA2, HTRA1-LoopA^HTRA3 and HTRA1-LoopA^HTRA4. **h** Alignment of partial protease domain sequences of HTRA family members highlighting the poor conservation of LoopA. Red stars indicate residues important for Fab interaction (see Supplementary Tables 4 and 5). Phylogram on the left based on sequence conservation of the HTRA1-4 protease domains. Boundaries of the protease domains are indicated on the left. SPR kinetic data in (**a, e–g**) are presented as the mean ± S.D. of three independent experiments.

activity of the full-length version of HTRA1 we generated mutants of HTRA1^FL in which LoopA was deleted (HTRA1^FL-ΔLoopA) or replaced by LoopA of HTRA2 (HTRA1^FL-LoopA^HTRA2). Consistent with the results of the HTRA1^PD mutants, we observed a decrease in catalytic activity of these mutants in both the casein-BODIPY and in the DKK3 cleavage assays (Supplementary Fig. 9a, b). As expected, Fab15H6.v4 no longer bound to these full-length mutants (Supplementary Fig. 9c, Supplementary Table 7) and, thus, was unable to inhibit their residual activities (Supplementary Fig. 9a, b). We also built a model of Fab15H6.v4 bound to full length HTRA1 using the previously determined low resolution SAXS structure of HTRA1

(Supplementary Fig. 9d) in order to assess potential obstructions to Fab binding by the accessory domains. In this model neither the peripheral N-terminal IGFBP/Kazal-like domains, nor the C-terminal PDZ domain interfere with Fab15H6.v4 binding to LoopA (Supplementary Fig. 9e, f), in agreement with results showing Fab binding to HTRA1^FL and inhibition of its enzymatic activity (Supplementary Fig. 9a–c).

Collectively, our results show that LoopA is essential for the catalytic function of HTRA proteases. In HTRA1, the LoopA function is not confined to a specific LoopA residue but requires the complete LoopA amino acid sequence.

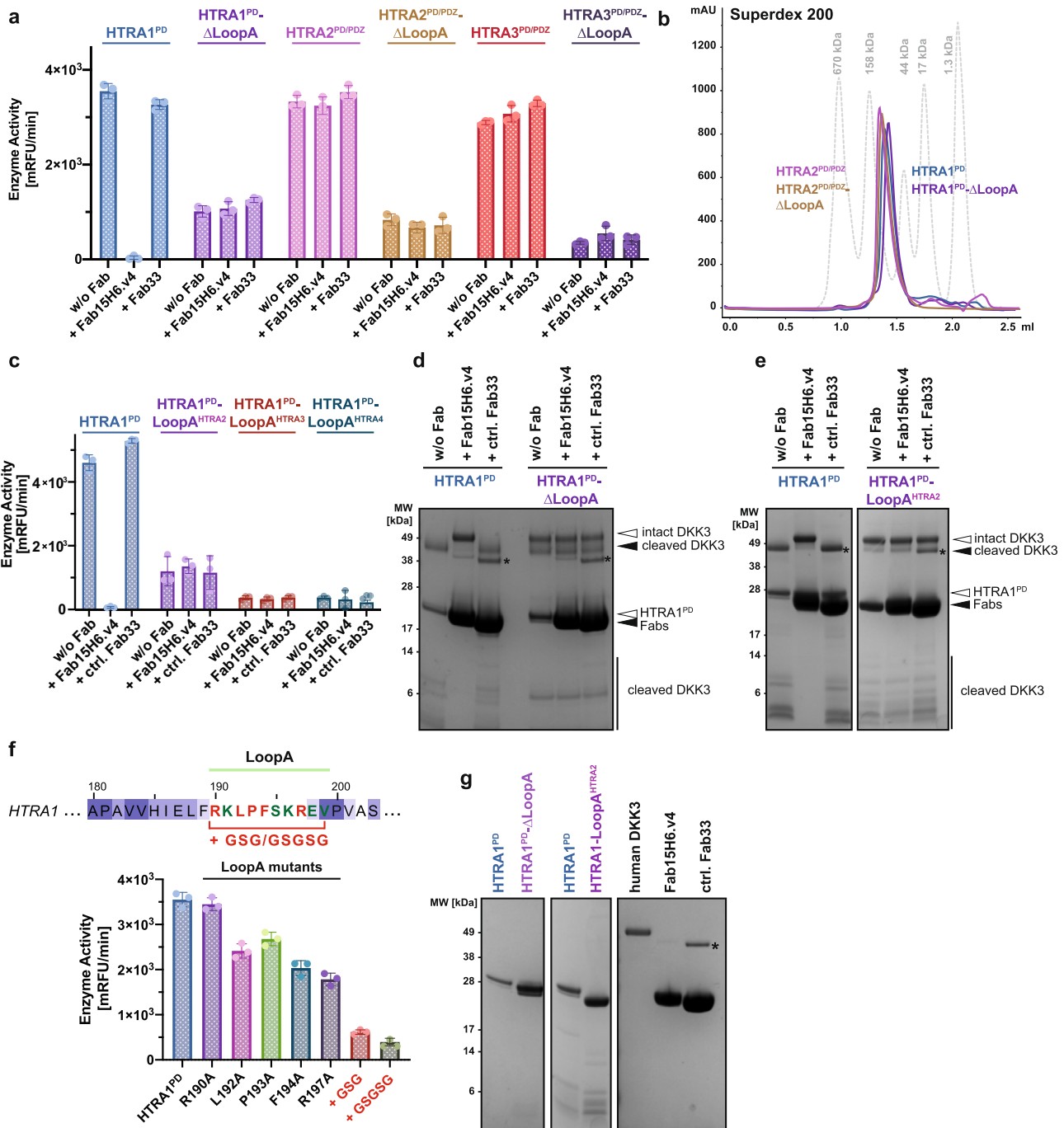

**Fig. 5 | Deletion or perturbation of LoopA diminishes catalytic activity within the HTRA family. a** Enzymatic activity of wildtype HTRA1$^{PD}$, HTRA2$^{PD/PDZ}$, and HTRA3$^{PD/PDZ}$ compared to their ΔLoopA mutants; Fab15H6.v4 and the control Fab33 did not modify the mutant activities. **b** Size-exclusion chromatography profiles of HTRA1$^{PD}$ (blue), HTRA1$^{PD}$-ΔLoopA (purple), HTRA2$^{PD/PDZ}$ (light purple) and HTRA2$^{PD/PDZ}$-ΔLoopA (brown) indicate formation of trimers (protein standards in grey). **c** Enzymatic activity of LoopA chimeras of HTRA1$^{PD}$ in the presence or absence of Fab15H6.v4. **d** In-vitro DKK3 cleavage assay using purified HTRA1$^{PD}$ or HTRA1$^{PD}$-ΔLoop mutant in the presence or absence of Fab15H6.v4. **e** In-vitro DKK3 cleavage assay using wildtype HTRA1$^{PD}$ and the LoopA swap chimera

HTRA1$^{PD}$-LoopA$^{HTRA2}$. **f** Design and enzymatic activity of single amino acid substitutions within LoopA of HTRA1$^{PD}$ and of GSG/GSGSG linkers. **g** Control gels showing individual proteins incubated under identical conditions as used for cleavage assays in **d**, **e** Asterisk (\*) in (**d**, **e**, **g**) indicates contaminant in the Fab33 preparations that overlaps with the cleaved DKK3 band. Bar graphs in **a**, **c**, **f** are presented as the mean ± S.D. of three independent experiments. Images in **d**, **e**, **g** as well as chromatogram in **b** are representative of two independent experiments. For experiments in **a**, **b** the HTRA2 and HTRA3 constructs comprised the protease and PDZ domains (HTRA2/3$^{PD/PDZ}$). Source data are provided as a Source Data file.

## Conservation of the conformational lock mechanism in the HTRA family

The finding that LoopA is important for the activities of HTRA proteases suggested the possibility that concordant to the HTRA1 paradigm, Fab binding to LoopA of other HTRA members may evoke their allosteric

inhibition. Because we did not have any antibodies to LoopA of other HTRA proteases available and since Fab15H6.v4 is HTRA1-specific, we decided to transfer the Fab15H6.v4 epitope to other HTRA proteases to test this hypothesis. For this we chose HTRA2 and HTRA3, which are the most distant and closest relatives to HTRA1, based on the sequence

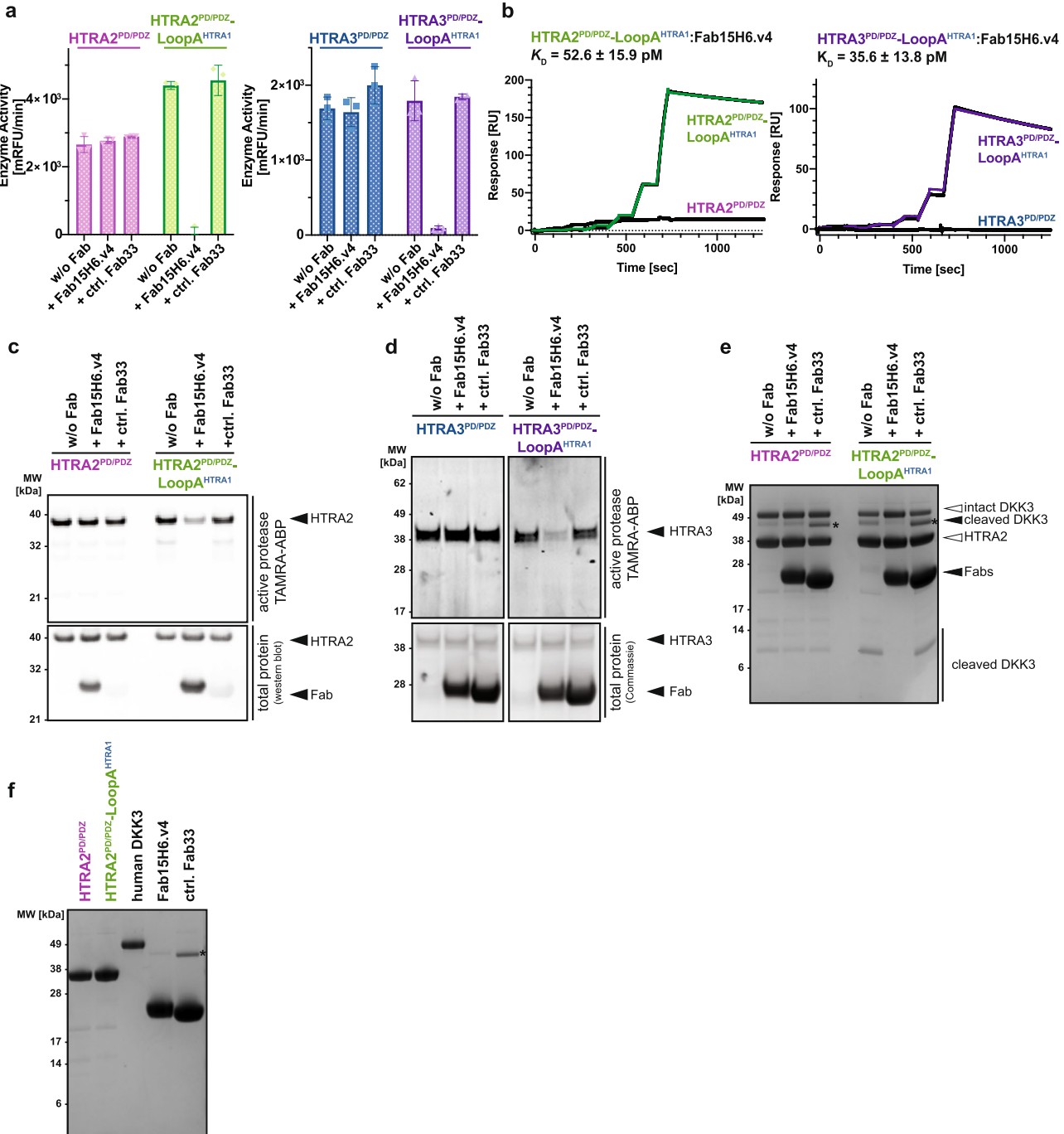

**Fig. 6 | Transfer of the Fab15H6.v4 allosteric inhibition mechanism to HTRA2 and HTRA3. a** Enzymatic activity of wildtype HTRA2$^{PD/PDZ}$ (pink) and HTRA3$^{PD/PDZ}$ (blue) compared to the chimeric HTRA2$^{PD/PDZ}$-LoopA$^{HTRA1}$ (green) and HTRA3$^{PD/PDZ}$-LoopA$^{HTRA1}$ (purple) proteins in the presence of Fab15H6.v4 or control Fab33. **b** SPR binding kinetics of Fab15H6.v4 interaction with wildtype HTRA2$^{PD/PDZ}$ and HTRA3$^{PD/PDZ}$ and their LoopA chimeras. **c, d** Labeling of HTRA2$^{PD/PDZ}$ and HTRA3$^{PD/PDZ}$ wildtype proteins compared to the LoopA chimeras using fluorescent activity-based probe (TAMRA-ABP) in the absence or presence of Fab15H6.v4 or control Fab33. **e** In-vitro cleavage of DKK3 substrate using wildtype HTRA2$^{PD/PDZ}$ or the chimeric HTRA2$^{PD/PDZ}$-LoopA$^{HTRA1}$ in the presence of Fab15H6.v4. No cleavage was detected using HTRA3$^{PD/PDZ}$ or HTRA3$^{PD/PDZ}$-LoopA$^{HTRA1}$ (not shown) **f** Control gel showing individual proteins incubated under identical conditions as used for cleavage assay in **e** Asterisks (*) in (**e**, **f**) indicate contaminant in the Fab33 preparation that overlaps with the cleaved DKK3 band. Bar graphs in **a** and kinetic data in **b** are presented as the mean ± S.D. of three independent experiments. Images in **c–f** are representative of two independent experiments. For all experiments in Fig. 6 the HTRA2 and HTRA3 constructs comprised the protease and PDZ domains (HTRA2/3$^{PD/PDZ}$). Source data are provided as a Source Data file.

conservation within the protease domain (Fig. 4h[9]). We generated the HTRA2$^{PD/PDZ}$-LoopA$^{HTRA1}$ and the HTRA3$^{PD/PDZ}$-LoopA$^{HTRA1}$ chimeric proteins, in which we replaced LoopA of HTRA2 or HTRA3 with that of HTRA1. Assays with casein-BODIPY substrate demonstrated that the LoopA changes did not impair enzyme activities and even slightly improved it in the case of HTRA2$^{PD/PDZ}$-LoopA$^{HTRA1}$ (Fig. 6a). A

superposition of the HTRA1$^{PD}$:Fab15H6.v4 structure with the crystal structures of HTRA2 (pdb 1LCY,[41]) and HTRA3 (pdb 4RIO,[42]) showed that in case of HTRA2 the C-terminal PDZ domain does not interfere with Fab binding (Supplementary Fig. 10). In HTRA3, the PDZ domain is located closely to the LoopA epitope, but the flexible linker region connecting the protease and PDZ domains may enable repositioning of the PDZ

domain to allow for unobstructed Fab binding (Supplementary Fig. 10). In agreement, SPR assays clearly demonstrated that engraftment of the HTRA1-LoopA onto HTRA2 and HTRA3 provided high-affinity binding for Fab15H6.v4, comparable to wildtype HTRA1[PD] (Fig. 6b), while wild-type HTRA2 and HTRA3 were not recognized. Remarkably, both the HTRA2[PD/PDZ]-LoopA[HTRA1] and the HTRA3[PD/PDZ]-LoopA[HTRA1] chimeras were fully inhibited by Fab15H6.v4 in the casein-BODIPY enzyme assay (Fig. 6a). Moreover, Fab15H6.v4 also inhibited labeling of HTRA2[PD/PDZ]-LoopA[HTRA1] and HTRA3[PD/PDZ]-LoopA[HTRA1] by the small fluorescent activity-based probe, although the inhibition of HTRA2[PD/PDZ]-LoopA[HTRA1] was not complete in this assay (Fig. 6c, d). We then tested both chimeric proteins in the in-vitro DKK3 cleavage assay. Wildtype HTRA2[PD/PDZ] and HTRA2[PD/PDZ]-LoopA[HTRA1] were able to cleave DKK3, albeit with very low activity (Fig. 6e). Addition of Fab15H6.v4 completely inhibited DKK3 cleavage by the HTRA2[PD/PDZ]-LoopA[HTRA1] chimera, but not by wildtype HTRA2[PD/PDZ] (Fig. 6e), in agreement with the casein-BODIPY assay results. We did not observe any cleavage of DKK3 by HTRA3[PD/PDZ] or by HTRA3[PD/PDZ]-LoopA[HTRA1] and, therefore, were unable to address Fab15H6.v4 inhibition in this assay.

In conclusion, these findings suggest that the inhibitory mechanism by which Fab15H6.v4 inhibits the HTRA2 and HTRA3 chimeric proteases is akin to the conformational lock mechanism described for HTRA1, and that allosteric modulation of protease activity by a LoopA-specific antibody should be achievable for other members of the HTRA family.

## Discussion

Inhibition of the serine protease HTRA1 by the clinical Fab15H6.v4.D221 is currently being explored in a Phase II clinical trial as a novel treatment option for GA (NCT03972709). However, the precise inhibition mechanism of the Fab remained puzzling because biochemical studies indicated that it neither dissociated the HTRA1 trimer nor bound to the active site. Herein we describe a long-range allosteric lock mechanism that inactivates HTRA1 catalysis as a result of Fab binding to the exposed LoopA, which was subsequently identified as being essential for the enzyme activity of HTRA family proteases.

Our cryo-EM structures show that the Fab-bound HTRA1 is locked in a non-competent conformation, in which structural rearrangements in the active site region, including Loop1 and Loop2, render the protease domain enzymatically inactive. Importantly, in contrast to the uninhibited HTRA1[23], this Fab-bound non-competent HTRA1 is no longer able to transition to the active state, as evidenced by the absence of conformational heterogeneity in the active site region in cryo-EM structures. MD simulations provided further evidence that in the Fab complex the two hallmark residues H220 and L345 are both trapped in non-productive conformations that abolish catalysis. This allosteric lock mechanism is consistent with the conformational selection model in that Fab15H6.v4 binding shifts the equilibrium from the competent to the non-competent state and thereby prevents substrates from sampling the active conformation. Consequently, Fab-bound HTRA1 is no longer able to process any substrates, as shown by functional cleavage assays in-vitro and was also observed in-vivo in a clinical Phase I study, in which intravitreal administration of the Fab15H6.v4.D221 completely blocked HTRA1-mediated cleavage of the biomarker DKK3 in the human aqueous humor samples of patients with GA secondary to AMD[31,33].

The long-range allostery of Fab15H6.v4 is based on the unique, extended architecture of HTRA1-LoopA and is distinct from other antibody- and peptide-mediated inhibition mechanisms for trypsin-like serine proteases[27–29,43–47]. Severe antibody-induced changes of the central active site Loop1 and Loop2 have been reported for the serine proteases urokinase-type plasminogen activator (uPA) and hepsin[46,48,49]. However, the allosteric inhibition mechanisms of the anti-uPA and anti-hepsin antibodies involve epitopes that are located much closer to the active site as compared to the LoopA epitope of HTRA1.

Importantly, these antibodies induce conformational disorder in the active sites[46,48,49], rather than a conformational lock as observed in our studies.

For therapeutic inhibition of proteases, targeting regions of low conservation, such as the LoopA of HTRA1, has several advantages over targeting the conserved active site. First, the high conservation of the active site within the HTRA family hampers the development of specific small molecule inhibitors that are able to discriminate between the different family members[26] and non-specific inhibitors may produce unwanted off-target effects. Second, as shown recently[50], sub-stochiometric inhibition of the catalytic center of HTRA1 using small molecules or peptides can lead to a partial activation of HTRA1 through a substrate-induced mechanism. Therefore, targeting less conserved allosteric sites, such as LoopA, distant from the catalytic center allows full inhibition without partial activation of HTRA1's enzymatic activity.

Like HTRA1, the other HTRA family members exhibit conformational plasticity within the catalytic center[51–53], and might be susceptible to a LoopA-directed allosteric lock mechanism akin to HTRA1. This hypothesis was tested by the use of chimeric HTRA2 and HTRA3 proteases containing the engrafted LoopA from HTRA1, which provided high-affinity Fab binding to the chimeras. We observed complete enzyme inhibition and reduced TAMRA-ABP labeling of these chimeric proteins by Fab15H6.v4, suggesting a factual transfer of the allosteric inhibition mechanism and confirming the hypothesis. A caveat concerning this conclusion is that it is based on the assumption that the transferred HTRA1-LoopA served as a true functional replacement of the native LoopA sequences of HTRA2 and HTRA3. Nevertheless, the results raise the intriguing possibility that antibodies selected for binding to the natural LoopA of HTRA2-4 proteases could effectively inhibit their enzyme activities by the same conformational lock mechanism as observed for Fab15H6.v4 and HTRA1. The finding that Fab binding to LoopA abolishes the activity of HTRA proteases suggested that LoopA is important for catalytic activity. This was indeed the case, as exemplified by LoopA deletion experiments, showing a reduction of enzyme activities of the HTRA1-3 mutants by more than 70%. Additional experiments with LoopA chimeras, single amino acid mutations, and LoopA linker insertions revealed a remarkable precision of the LoopA function, in that optimal enzyme activity, requires the exact natural LoopA amino acid sequence and an unaltered LoopA connectivity. However, it remains currently unclear how the allosteric signal is transmitted from the distant LoopA to the catalytic center. The importance of LoopA for HTRA catalytic activity raises the possibility that potential interactors of LoopA could regulate HTRA protease activity in the local environment. The regulation of HTRA protease activity is particularly pertinent considering that the secreted HTRA members (HTRA1, 3, 4) do not follow the zymogen activation paradigm of trypsin fold serine proteases but are secreted as trimeric active enzymes not requiring activation cleavage.

In conclusion, our study provides a detailed insight into the mode of inhibition of the trimeric serine protease HTRA1 by a clinical Fab fragment. Together with the reported safety, tolerability, and pharmacodynamics of Fab15H6.v4.D221[31–33], our findings provide an improved understanding of this Fab-based therapeutic approach for the treatment of GA. The therapeutic potential of Fab15H6.v4.D221 may not be limited to AMD, as HTRA1 has been implicated in other chronic diseases, such as osteoarthritis[54,55]. HTRA1 expression is increased in synovial fluid of osteoarthritis patients and may contribute to cartilage degradation[18,56]. Specific inhibition of HTRA1 activity could protect the integrity of the pericellular matrix[54] and may serve as a novel therapeutic opportunity for the treatment of arthritic diseases. In contrast, the role of HTRA1 in CARASIL[13,21,22], certain cancers[57,58], and in Alzheimer's disease[19,59] may require different therapeutic approaches aimed at enhancing, rather than inhibiting HTRA1 activity.

## Methods

### Constructs of HTRA1-4 and their mutant forms

Human HTRA1 protease domain (HTRA1[PD]; D161-K379, 24 kDa), the HTRA2 protease/PDZ domain (HTRA2[PD/PDZ]; A134-E458), the HTRA3 protease/PDZ domain (HTRA3[PD/PDZ]; L130-M453) and their LoopA mutants, as well as mouse HTRA1 protease domain (mouse HTRA1[PD]; D161-K379, 24 kDa) were cloned into N-terminal TEV or thrombin cleavable His$_6$ tag vector for *E. coli* expression. HTRA1[FL] (Q23-P480, 51 kDa) and its LoopA mutants were cloned into N-terminal TEV cleavable His$_6$ tag vector for insect cell expression. LoopA swap mutants and chimeric proteins were generated by taking the LoopA sequence of HTRA1 (R190-V199) and replacing it with the LoopA sequences of HTRA2 (D168-V177) or HTRA3 (L161-V170) or vice versa. Deletion of LoopA mutants were generated by replacing LoopA sequences with a beta-hairpin motif[40].

### Expression and purification of recombinant proteins and Fab constructs

HTRA1-3 proteins containing PD or PD/PDZ domains were expressed in *E. coli* and purified as previously described[23,31]. Full length HTRA1 (HTRA1[FL]) proteins were expressed in *Trichoplusia ni* cells. *E. coli* lysates or insect cell media were loaded onto Ni-NTA column (Qiagen) and washed with 10 column volumes of wash buffer (500 mM NaCl, 50 mM Tris pH 8.0, 20 mM imidazole). HTRA1 proteins were eluted with elution buffer (200 mM NaCl, 50 mM Tris pH 8.0, 300 mM imidazole) and further purified by size exclusion chromatography on a Superdex 200 column (GE Healthcare) with 200 mM NaCl, 50 mM Tris pH 8.0, 0.25% CHAPS. Peak fractions corresponding to HTRA trimers were collected. Protein purity was verified by SDS-PAGE (Supplementary Fig. 11) and LC/MS.

The Fab15H6.v4 DNA clone[31] was expressed in 500 ml of Complete CRAP media with 100 μg/ml carbenicillin (1:100 dilution) in Ultra Yield (Thomson) 2 l flasks and incubated for 36 h at 30 °C on an orbital shaker (Infors HT, Multitron) at 200 rpm. Cells were harvested by centrifugation. The resulting cell paste was stored frozen at −80 °C. Cell paste was thawed into PBS, 25 mM EDTA, 1 mM PMSF/gram of cell paste. The mixture was homogenized and then passed twice through a microfluidizer (Microfluidics). The suspension was centrifuged and clarified supernatant was filtered through a 0.8-micron Nalgene filter and loaded onto a Protein G column (GE) equilibrated with PBS at a flow rate of 5 ml/min. The column was washed with PBS buffer to base line and then eluted with 0.6% v/v acetic acid in distilled water. Peak fractions were pooled and loaded onto a SP Sepharose column (GE Healthcare) equilibrated in 25 mM MES pH 5.5. The column was washed with 2 column volumes of wash buffer (25 mM MES pH 5.5). A linear gradient (0–0.5 M NaCl, 25 mM MES pH 5.5) over 10 column volumes was applied to elute the protein with a flow rate of 5 ml/min. 2 ml fractions were collected and peak fractions corresponding to Fab15H6.v4 were pooled and concentrated to 10 mg/ml using a spin concentrator. Protein purity was verified by SDS-PAGE and LC/MS.

Single chain Fv (scFv) fragment of Fab15H6.v4 was generated by linking the variable domain of the light chain (D1-K106) with the variable domain of the heavy chain (E1-S119) by a (GGGGS)$_4$ linker and fusing it onto a human IgG1 Fc sequence. The construct was expressed in CHO cells and secreted as scFv-Fc fusion construct. CHO cell medium was cleared and loaded onto a MabSelect Sure column (GE Healthcare) equilibrated with 150 mM NaCl, 25 mM Tris pH7.5, 5 mM EDTA and washed with 400 mM K$_3$PO$_4$, 25 mM Tris pH7.5, 5 mM EDTA, 0.2% P20. The fusion construct was eluted with 150 mM NaCl, 50 mM NaCitrate pH 3.0 and neutralized immediately. Peak fractions were collected and further purified with Superdex 200 column (GE Healthcare). The scFv-Fc fusion construct was cleaved with LysC (Promega) in PBS at 4 °C overnight. The mixture of scFv and Fc was applied to a MabSure Select column to capture the Fc, while the flowthrough containing the scFv was further purified on a Superdex 75

column (GE Healthcare) equilibrated in 250 mM NaCl, 25 mM Tris pH 8.0. Protein purity of the scFv peak fraction was verified using SDS-PAGE and LC/MS.

Size exclusion profiles of HTRA proteins and their LoopA deletion mutants were generated using a Superdex 200 Increase 3.2/300 column (GE Healthcare) equilibrated with 200 mM NaCl, 20 mM HEPES pH7.5, 0.25% CHAPS. For size estimation molecular weight gel filtration standard (grey, Bio-Rad Laboratories) was applied using the same conditions.

Recombinant HTRA1 substrates DKK3 (Order #: 1118-DK), BIGLYCAN (2667-CM), DECORIN (143-DE) were purchased from R&D Systems.

### Crystallography of Fab15H6.v4 and LoopA peptide:Fab15H6.v4 complex

Crystallization trials of Fab15H6.v4 were performed with high-throughput commercial screens by Qiagen and Hampton using a Mosquito liquid handler (SPT Labtech). Optimized crystal conditions were 2.1 M ammonium sulfate, 0.1 M Tri-sodium citrate pH 5.1 at 18 °C using sitting drops. Crystals were cryoprotected with 25% (v/v) glycerol added to the reservoir solution and immersed in liquid nitrogen. Data was collected at SSRL 12-2 and processed using XDS. Molecular replacement was performed using Phaser[60] and anti-HER2 Fab (pdb 1FVD[61]) as the search probe. Molecular replacement found 1 fab/asymmetric unit. Refinement was performed using Phenix.refine with manual model building using Coot. Final refinement was performed using Buster[62].

The HTRA1-LoopA peptide:Fab15H6.v4 complex was prepared by adding 2 mg of peptide ([190]RKLPFSKREVPV[201], NeoBiolab, Cambridge, Massachusetts, USA) to 1 ml of Fab15H6.v4 at 10 mg/ml protein concentration and incubating overnight at 4 °C. High-throughput screens were set up using 0.2 μl complex + 0.2 μl reservoir in sitting drops at 18°C. After 2 months, rod shaped crystals grew from 2.2 M ammonium sulfate, 0.1 M potassium acetate (Qiagen, AmSO$_4$ suite). The crystals were cryoprotected with 25% (v/v) glycerol added to the reservoir solution and immersed in liquid nitrogen. Data was collected at SSRL 12-2 and processed using XDS[63]. The solved structure of Fab15H6.v4 was used as the molecular replacement search probe in Phaser[60] with 1 complex found per asymmetric unit. Refinement was performed using Phenix.refine with manual fitting in Coot. The final model was refined with Buster[62]. 2mFo-DFc omit maps contoured at 1.0 σ for the apo Fab and the LoopA peptide:Fab15H6.v4 structures are shown in Supplementary Fig. 12.

### Cryo-EM sample preparation, data collection and processing

HTRA1[PD] or HTRA1[PD/SA] proteins were mixed with Fab15H6.v4 in a 1:2 molar ratio and incubated on ice for 30 min. The mixture was then injected onto a Superdex 200 Increase 3.2/300 column (GE Healthcare, Figs. 1d and S3a) equilibrated with 200 mM NaCl, 20 mM HEPES pH 7.5, 0.25% CHAPS. Peak fractions corresponding to the HTRA1:Fab complex were identified and collected (Figs. S2a and S3a). Au substrate Quantifoil (Quantifoil GMBH) cryo-EM grids (0.6/1.0) with 25 nm thick Au foil were incubated with a thiol-reactive, self-assembling reaction mixture of 4 mM monothiolalkane(C11)PEG6-OH (11-mercaptoundecyl) hexaethylenglycol (SPT-0011P6, SensoPath Technologies, Inc., Bozeman, MT),[64]. Grids were incubated with this self-assembled monolayer (SAM) solution for 24 h. Prior to freezing, grids were removed from the SAM solution and rinsed with ethanol. Then 3 μl of HTRA1:Fab complex was applied to SAM, 25 nm Au foil grids. After sample application, grids were then blotted for 3.5 seconds and plunged into liquid ethane, using the Leica Microsystems automatic plunge freezer (EM GP2, Leica Microsystems, Buffalo Grove, IL). Movie stacks were collected using SerialEM[65] on a Titan Krios (Thermo Fisher Scientific) operated at 300 keV using a bioquantum energy filter equipped with a K3 Summit direct electron detector camera (Gatan Inc., Pleasanton, CA). Images were recorded in EFTEM mode at 105,000 × magnification corresponding to 0.838 Å per pixel, using a 20 eV energy

slit. Each image stack contains 60 frames recorded every 0.05 s for an accumulated dose of ~60 e Å − 2 and a total exposure time of 3 s. Images were recorded with a set defocus range of 0.5 to 1.5 μm.

## Cryo-EM structure determination of HTRA1:Fab15H6.v4 complexes

Image processing was performed using a combination of cisTEM[66], RELION 3.1[67] and cryoSPARC v3.2 (Structura Biotechnology Inc.,[68]). We solved a first structure using the catalytically inactivate HTRA1$^{PD/SA}$ complexed with Fab15H6.v4 (Supplementary Fig. 2). A total of 13,201 movies were collected and motion-corrected in cisTEM. Images were filtered based on the CTF fit resolution (better than 4.8 Å) and score (0.08 or better). From the 7,366 images thus selected, a total of 505,669 coordinates were picked using the circular blob picking tool within cisTEM using a radius of 60 Å. Particles were sorted in 2 rounds of 2D classification using cisTEM, yielding 219,676 particles, showing a distinct three-fold symmetry in top views (Supplementary Fig. 2b, c). The particles were used for ab-initio 3D reconstruction, auto and local refinement using five classes and assuming a C3 symmetry. Five maps were generated with resolutions between 3.9 Å and 4.25 Å. The best map (3.9 Å) and the corresponding particle parameters were selected and the full stack of 219,676 particles was polished using RELION Bayesian particle polishing. The polished particles were reimported into cisTEM for 3D refinement. The HTRA1$^{PD/SA}$:Fab15H6.v4 3D reconstruction, with an estimated resolution of 3.3 Å, was obtained after auto refinement in cisTEM, during which no data beyond 4.6 Å resolution was used for parameter refinement. This map was sharpened by applying a negative B-factor of −90 Å$^2$ from the origin to 12 Å, flattening the spectrum from 12 Å onwards, and applying a figure-of-merit filter. This sharpened map was used to build the model.

For the focused HTRA1$^{PD/SA}$:Fab15H6.v4 Fv map, symmetry expansion was applied to the 219,676 polished particles yielding 659,028 particles. A mask including the HTRA1 trimer and only one Fab-Fv was used to low-pass filter to 20 Å regions outside the mask during further cycles of refinement. Local refinement using data up to 4.6 Å yielded a map with an estimated global resolution of 3.6 Å where only one of the Fv domains was resolved. This final map was sharpened as described above and used to refine the atomic model within the catalytic center of HTRA1 and the LoopA epitope contacting Fab15H6.v4. Although having a worse global resolution, resolution within the Fv fragment was improved and allowed a better placement of the previously solved crystal structure of Fab15H6.v4.

For the active HTRA1$^{PD}$ in complex with Fab15H6.v4 (Fig. S3) we collected 7,002 movies and corrected them using motion correction in cisTEM. Images were selected based on detected CTF fit resolution of 4.8 Å or better and a score of 0.08 or better. A total of 5,345 images were used for particle picking using a circular blob of 60 Å radius, yielding 566,088 picks. Particles were sorted in 2 rounds of 2D classification using cisTEM, generating 184,782 particles. The particles were polished using RELION Bayesian polishing and imported into cryoSPARC. From the polished particles, we generated two maps applying C3 and no (C1) symmetry. After refinement in cryoSPARC, the C3 symmetry map resulted in a final map with 3.3 Å resolution and the C1 map with a resolution of 3.6 Å.

## Model building, structural analysis, and visualization

The HTRA1$^{PD/SA}$ trimer (pdb 3TJO) and the solved apo Fab15H6.v4 crystal structure were fitted as rigid body into the cryo-EM maps. After manual inspection and adjustment, multiple rounds of real space refinement using phenix.real_space_refinement[69] were used to refine the initial models. Ramachandran restraints were used during model building and refinement. The model was further interactively adjusted in Coot[70] and finally validated using phenix.validation_cryo-EM[69] with built-in MolProbity scoring[71]. Final statistics for the structural models are summarized in Supplementary Table 2. Representative sections highlighting the quality of the fit for both cryo-EM maps and models are shown in Supplementary Figs. 2i and 3j.

Figures visualizing cryo-EM maps and structural models were generated using ChimeraX[72] and The PyMOL Molecular Graphics System, Version 2.0 (Schroedinger LLC). Electrostatic surface visualization was generated using APBS Electrostatic plugin within PyMOL 2.0.]. Positional conservation scores from multiple sequence alignments were obtained with the program ConSurf (v2016)[73].

## Molecular dynamics simulations

Molecular dynamics (MD) simulations were performed on protomers of HTRA1$^{PD}$ using models derived from the competent state (pdb 3TJN, 3.0 Å resolution, chain B and pdb 3NZI, 2.75 Å resolution, chain A, with active site inhibitor removed), and the Fab15H6.v4:HTRA1$^{PD}$ complex (3.3 Å resolution, chain A, with Fab removed). Prior to running MD simulations of the HTRA1 protomers, we performed several steps of refinement and modelling for each structure. Prime (Schrodinger) was used to build missing side chains and cap the protein termini with N-acetyl and N-methylamine groups. Rosetta Remodel was used to build in missing loops[74]. All titratable residues were left at their dominant protonation state at pH 7.0 except H220 which was protonated on the delta nitrogen and deprotonated on the epsilon nitrogen of H220. Each protomer was then solvated in an octahedral-shaped box using a TIP3P explicit water model with waters modeled 12 Å from the edge of the protomer. All simulations were prepared with 150 mM Na$^+$ and 150 mM Cl$^-$ ions and charges neutralized. Hydrogen mass repartitioning was performed on the protein atoms[75] and a simulation time step of 4 fs was used.

MD simulations were performed on GPU's using the CUDA-enabled version of PMEMD in Amber[76]. Each protomer was first minimized using conjugate-gradient energy minimization for 2,000 steps with a harmonic restraining potential force constant of 10 kcal/mol/Å$^2$ applied to the solute atoms. The systems were then heated to 300 K at a constant pressure of 1 atmosphere with 10 kcal/mol/Å$^2$ harmonic restraints placed on the protein atoms. Next, another step of conjugate-gradient energy minimization was run for 2000 steps without restraints. The system was then equilibrated for 1 ns with a restraint force constant of 1 kcal/mol/Å$^2$. Following equilibration, restraints were removed for the production simulations using the NPT ensemble and the Langevin thermostat for 1 μs with all bond lengths to hydrogens constrained using the SHAKE algorithm[77]. Noncovalent interactions were cut off at 10 Å and Particle Mesh Ewald electrostatics were calculated for long-range interactions. For all other simulation parameters, default values were used. The protocol described above was run 3 times independently for each system to generate a total of 3 μs of data for each system. We specifically monitored the distance between L345 to the catalytic S328 (L345-C$^{\gamma1}$ and S328-O$^{\gamma}$, Figs. 3a, b, d, S6a) and the distance between H220 and the catalytic S328 increases (H220-N$^{\epsilon2}$ and S328-O$^{\gamma}$, Figs. 3a, c, e, S6b) during the MD simulations. The measured distances throughout the simulations were plotted and the expected distances for competent and non-competent conformations were indicated in these plots as guides (Figs. 3b–e, S6a, b). Differences in first, second and third derivative were visualized by PCA analysis (Fig. 3f, Supplementary Fig. 6c, d), and variance was plotted according to its Eigenvalue (Supplementary Fig. 6e). We also monitored the overall flexibility of each individual residue during the MD simulations (Supplementary Fig. 6f). Analysis of each simulation was carried out using VMD[78], CPPTRAJ[79], and Bio3D[80].

## Measurement of binding affinities by SPR analysis

All binding affinities of Fab15H6.v4 and HTRA1 proteins were determined using a Biacore S200 instrument (GE Healthcare) at 25 °C and with HBS-EP (100 mM HEPES pH7.5, 150 mM NaCl, 3 mM EDTA, 0.005% surfactant P20) as running buffer. All kinetics measurements were performed using single-cycle kinetics and referenced by subtracting

the signal to the blank flow cell. Referenced datasets that showed binding were fitted to a 1:1 Langmuir binding model using the GE Biacore S200 instrument software (GE Healthcare). Final kinetics and standard deviations were calculated based on at least three independent experiments.

Binding kinetics of scFv15H6.v4, Fab15H6.v4 and its mutants were measured by immobilization of His$_6$-tagged HTRA1$^{PD}$ using the His-capture kit (GE Healthcare). Approximately 200 RUs of wildtype His$_6$-tagged HTRA1$^{PD}$ were captured on the CM5 chip. The Fabs and scFv15H6.v4 were analyzed using single-cycle kinetics in a three-fold dilution series starting from 10 nM or 1 µM final concentrations. The absence of binding was confirmed by injecting up to 3 µM final concentration of Fab15H6.v4 mutant.

Binding kinetics to Fab15H6.v4 of wildtype and mutant HTRA1$^{PD}$ (R190A, L192A, P193A, F194A, R197A, HTRA1$^{PD}$-ΔLoopA, HTRA1$^{PD}$-LoopA$^{HTRA2/3/4}$), HTRA1$^{PD}$ preincubated with ABP (biotin-DPMFKLV-phosphonate, 7-mer ABP) and mouse HTRA1$^{PD}$ were determined by immobilizing the His$_6$-tagged HTRA1 proteins using the His-capture kit (GE Healthcare) on a CM5 chip. The same capture protocol was used for HTRA1$^{FL}$ and its mutants, as well as for wildtype and mutant forms of HTRA2$^{PD/PDZ}$ and HTRA3$^{PD/PDZ}$. Approximately 200 RUs of HTRA proteins were captured. Affinities to Fab15H6.v4 was measured in single cycle kinetics mode using three-fold dilution series starting from 1 nM or 1 µM final concentrations. Absence of binding was further verified using 3 µM Fab concentrations.

Binding kinetics of synthetic peptides of the LoopA from HTRA1 (wildtype and five single amino acid mutants), from HTRA2, HTRA3, and HTRA4 (all wildtype sequences) were measured directly by immobilizing Fab15H6.v4 onto a CM5 chip using the Amine Coupling kit (GE Healthcare). 1000 RUs of Fab15H6.v4 were immobilized and affinities were measured by single cycle kinetics and a three-fold dilution series of peptides starting from 10 nM (HTRA1 peptides) or 3 µM (HTRA2, HTRA3 or HTRA4 peptides).

### Peptide synthesis

Peptides were custom synthesized by Abclonal (Woburn, MA) using standard fluorenylmethoxycarbonyl (Fmoc) chemistry. Crude peptides were purified on preparative HPLC to a purity of >95% as determined by analytical LC/MS.

| | |
|---|---|
| HTRA1 LoopA wildtype | $^{190}$RKLPFSKREVPV$^{201}$ |
| HTRA1 LoopA R190A | $^{190}$AKLPFSKREVPV$^{201}$ |
| HTRA1 LoopA L192A | $^{190}$RKAPFSKREVPV$^{201}$ |
| HTRA1 LoopA P193A | $^{190}$RKLAFSKREVPV$^{201}$ |
| HTRA1 LoopA F194A | $^{190}$RKLPASKREVPV$^{201}$ |
| HTRA1 LoopA R197A | $^{190}$RKLPFSKAEVPV$^{201}$ |
| HTRA2 LoopA wildtype | $^{168}$DRHPFLGREVPI$^{179}$ |
| HTRA3 LoopA wildtype | $^{161}$LRHPLFGRNVPL$^{172}$ |
| HTRA4 LoopA wildtype | $^{188}$GRLLHGSRLVPV$^{199}$ |

### Enzymatic assays

The indicated concentrations of HTRA proteins are those of the monomer and not of the trimer. Activity of HTRA1$^{FL}$, HTRA1$^{PD}$, and their mutant proteins were measured using EnzCheck Protease Assay Kit (Thermo Fisher Scientific, E6638) containing a fluorescence-quenched casein-BODIPY conjugate in HTRA1 cleavage buffer (200 mM NaCl, 50 mM Tris pH 8.0, 0.25% CHAPS). HTRA1 proteins were mixed with Fab15H6.v4 or control Fab33 (human anti-PCSK9 Fab, prepared as described[35]), incubated at 37 °C for 5 min and casein substrate was added. The final concentration in the reaction mixture were: 30 nM of mouse or human HTRA1$^{PD}$, HTRA1$^{FL}$ or LoopA mutants, 1 µM Fab15H6.v4 or control Fab33 and 50 µM casein substrate.

Cleavage of casein-BODIPY was measured at 37 °C during a 10 min time period using a SPECTRAmax M5 microplate reader (Molecular Devices) and the linear rates of substrate cleavage were determined. All measurements were run in triplicate and normalized to baseline control activity lacking HTRA1 enzyme. The results presented are the mean ± S.D. of at least three independent experiments. The concentrations of HTRA2$^{PD/PDZ}$, HTRA3$^{PD/PDZ}$, and their LoopA mutants were 300 nM in the otherwise identical reaction mixture. Titration of 10 nM HTRA1$^{PD}$ was performed with casein-BODIPY substrate and decreasing concentration of Fab15H6.v4 (32 nM to 6.5 nM).

DKK3, BIGLYCAN, and DECORIN cleavage assays were performed using recombinant proteins in the presence of purified HTRA proteins with and without Fab15H6.v4, scFv15H6.v4 or control Fab33. 4 µM of DKK3, BIGLYCAN, or DECORIN were incubated with 1 µM HTRA proteases (HTRA1$^{PD}$, HTRA1$^{FL}$, HTRA2$^{PD/PDZ}$, HTRA3$^{PD/PDZ}$, and their LoopA mutants) in cleavage buffer and in the presence or absence of 10 µM Fab15H6.v4 or Fab33 at 37 °C overnight. Reduced samples were loaded onto 12% NuPAGE Gel (Thermo Fisher Scientific), run in 1x MES SDS Running buffer (Thermo Fisher Scientific) and stained with SimplyBlue SafeStain (Thermo Fisher Scientific) according to manufactures protocol. Gels were imaged on a GelDoc Imager System (Bio-Rad).

### Assays with activity-based probes

Labeling of HTRA1$^{FL}$, HTRA1$^{PD}$, HTRA2$^{PD/PDZ}$, HTRA3$^{PD/PDZ,}$ and their LoopA mutants with the fluorescent TAMRA activity-based probe (TAMRA-ABP) was performed as previously described[31]. 1 µM of HTRA proteins were incubated with 5 µM Fab15H6.v4 or control Fab33 for 1 h at room temperature in cleavage buffer, and then 10 µM TAMRA-ABP was added and further incubated for 1 h at room temperature. Reduced samples were loaded onto a 12% NuPAGE Gel (Thermo Fisher Scientific), run in 1x MES SDS Running buffer (Thermo Fisher Scientific), and fluorescence was measured using a Typhoon TRIO variable-mode imager or iBright imager (GE Healthcare). Gels were either stained with SimplyBlue SafeStain (Thermo Fisher Scientific), or the gels were blotted onto a nitrocellulose membrane using the iBlot Transfer System (Thermo Fisher Scientific) and total HTRA protein levels were detected by standard western blotting using biotinylated anti-HTRA1/HTRA2 19G10 mouse IgG2a monoclonal antibody (0.1 µg/ml Genentech,[31]) and Streptavidin-Poly-HRP (1:20.000, GE Healthcare). The same protocol was used to detect labeling in presence of the scFv15H6.v4 and mouse HTRA1$^{PD}$.

The DPMFKLV-phosphonate ABP (7-mer ABP) was synthesized using the previously described methodology[31]. The final product was purified by prep-HPLC (neutral condition) and purity was assessed by LC/MS. HTRA1$^{PD}$ was incubated with 7-mer ABP (10x molar excess) for 1 h at room temperature. The reaction mixture was then applied to a Superdex75 size exclusion column to remove excess 7-mer ABP and the purified covalent complex of HTRA1$^{PD}$ and 7-mer ABP used for SPR binding assays (Fig. 1e).

### Reporting summary

Further information on research design is available in the Nature Research Reporting Summary linked to this article.

## Data availability

The data that support this study are available from the corresponding authors upon reasonable request. The EM maps have been deposited in the Electron Microscopy Data Bank (EMDB) under the accession codes EMD-25163 (HTRA1$^{PD/SA}$:Fab15H6.v4 complex) and EMD-25162 (HTRA1$^{PD}$:Fab15H6.v4 complex). The atomic coordinates have been deposited in the Protein Data Bank (PDB) with the accession codes 7SJO (HTRA1$^{PD/SA}$:Fab15H6.v4 complex), 7SJN (HTRA1$^{PD}$:Fab15H6.v4 complex), 7SJM (apo-Fab15H6.v4) and 7SJP (LoopA peptide:Fab15H6.v4 complex). Other structures used as alignments for illustration are available in the PDB, including 3NZI, 3TJO and

3TJN. All reagents are available from the lead contact under a material transfer agreement with Genentech. Source data underlying Figs. 1a–c, 5a, c–g, 6a, c–f, Supplementary Figs. 1a–d, g, h, 2a, 3a, 8b, c, e, 9a, b, 11a, b, and Supplementary Tables 1, 4–6 are provided as a Source Data file. Source data are provided with this paper.

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

## Acknowledgements

We would like to thank Aimin Song and Jeffrey Tom for synthesis of peptides, the BMR group at Genentech for help with construct design and expression of proteins and Saeed Izadi for support and input on the MD simulations. We thank Bob Lazarus and Claudio Ciferri for helpful discussions on the manuscript, Andrea Cochran for advice on the β-hairpin motif construct and Dick Vandlen for preparing the scFv15H6.v4. Use of the Stanford Synchrotron Radiation Lightsource, SLAC National Accelerator Laboratory, is supported by the U.S. Department of Energy, Office of Science, Office of Basic Energy Sciences under Contract No. DE-AC02-76SF00515. The SSRL Structural Molecular Biology Program is supported by the DOE Office of Biological and Environmental Research, and by the National Institutes of Health, National Institute of General Medical Sciences (P30GM133894).

## Author contributions

S.G., M.U., J.H., W.L., and D.K. designed the experiments. S.G., M.U., W.T., E.G., J.H., W.L., A.E., C.A., I.T., A.R., and D.K. performed experiments and/or analyzed data. D.K. and S.G. conceived and supervised the project. S.G. and D.K. wrote the manuscript with input from all authors.

## Competing interests

All authors were employees of Genentech Inc., a for-profit institution, at the time when the studies were performed.
