## [Peer Review File · Nature Communications]

Allosteric inhibition of HTRA1 activity by a "conformational lock" mechanism to treat age-related macular degenerationReviewers' Comments:

Reviewer #1:

Remarks to the Author:

The manuscript titled : "Allosteric inhibition of HtrA1 activity by a "conformational lock" mechanism to treat age related macular degeneration" by Stefan Gerhardy et al is a detailed, comprehensive study of regulation (inhibition) of the HtrA1 protease by the Fab15H6.v4 fragment of the anti-HtrA1 antibody, binding to the HtrA1 protease domain (PD).

The study is justified by the fact that HtrA1 is known to be associated with age related macular degeneration (AMD); AMD can be categorized into neovascular AMD and geographic atrophy (GA) and there is no treatment available for GA. A slightly modified version of the Fab15H6.v4, namely the Fab15H6.v4.D226, is currently being investigated as a novel treatment for GA in a Phase II clinical trial. Since the molecular basis of the HtrA1 activity inhibition by the Fab15H6.v4 has not been investigated in detail, this study is an important step towards understanding the mechanism of action of the clinically used Fab.

The Authors performed a series of experiments including cryogenic electron microscopy (cryoEM), X-ray crystallography, molecular dynamics simulations and biochemical assays and showed that the Fab binds to a small epitope contained in the LA loop of HtrA1 PD, at a comparatively long distance (about 30 Å) from the serine protease active center. They solved the structure of the HtrA1-Fab complex at a high resolution, identifying the amino acid residues involved in binding on both sides. They proved that the LA loop is important for activity regulation and showed that the Fab binding locks the HtrA1 active site (including the loops L1, L2, LD and residues H220, L345) in an inactive conformation - the active center of the Fab-bound HtrA1 is unable to bind a substrate and cleave it. The Fab binding was specific to the HtrA1 LA loop as it did not interact with chimeric HtrA1 carrying the LA loops of HtrA2-4. They also proved that Loop A regulates other HtrA human proteases (HtrA2/3), furthermore transfer of the HtrA1 LoopA to HtrA2 and HtrA3 enabled the Fab to inhibit the resulting chimeric proteases.

The Authors concluded that the long range allosteric mechanism by which Fab15H6.v4 inhibits the HtrA2 and HtrA3 chimeric proteases is similar to the conformational lock mechanism described for HtrA1. They also postulated that allosteric modulation of protease activity by a LoopA-specific antibody should be achievable for other members of the HtrA family. I believe this is probable and such approach could be used in future, should inactivation of any of the HtrAs be beneficial from the medical point of view.

In my opinion, the experimental work has been done exceptionally well, the techniques are state of the art, each conclusion has been based on two or more experimental approaches, all conclusions are well based on the experimental data.

The findings are novel and important for better understanding the regulation of human HtrA proteases, and are significant from the medical point of view.

The results are well and clearly illustrated, the paper is very well written.

However, I have some comments.

1. The placing of control gels showing the proteins used in the experiments demonstrating substrate cleavage assays in the Supplementary Figure 10 may have saved some space but it is a most unusual practice. It is hard to believe that the protein gels run separately as controls are identical with the experimental gels shown in the figures to which the S10 refers. It is very inconvenient to find an appropriate control in a different figure, especially in a situation when there is no information that such control exists. The controls should be shown together with the cleavage assays in Figs 1b, 5de, 6e. Besides, the legend to Fig. S10 is confusing: (1) it refers to Fig.5g – there is no Fig5g; (2) it refers to Fig. 7 – there is no Fig. 7; (3) the gel shown in S10d does not correspond to Fig5f, as stated; (4) one of the lanes in S10e is not described.
2. It would help to better understand the results if the molecular weights of the HtrA1PD, HtrA1PD-PDZ and HtrA1FL were stated in the text – the first chapter of Results would be a good place.
3. The cleavage of the HtrA1 specific substrates (i.e. DKK3, decorin, biglycan) is highly inefficient compared to the casein substrate processing (considering enzyme/substrate ratio and incubation time). I think the Authors should comment on this.

4. The Authors showed that the full length HtrA1 and HtrA1-PD was inhibited by the Fab, and that the chimeric HtrA2-PD and HtrA3-PD carrying the LA loops of HtrA1 were also inhibited by the Fab. They postulated that the HtrA2/HtrA3 could be inhibited by specific Fabs interacting with their LA loops. However, the naturally existing HtrA2/3 have PDZ domains which are positioned differently with respect to PD (including LA loops) compared to HtrA1 - this is especially visible in HtrA2. Did the Authors try to model the Fab- HtrA2/3 interactions to see if there is no steric hindrance due to the different position of PDZ? They should comment on this in the paper.

5. Methods: Please, state the time range in which the cleavage of the Bodipy-casein was recorded.

6. In the Discussion, the structural and kinetic data concerning the HtrA1 inhibition are discussed in-depth. Contrary to this, the possible medical applications of the proposed general mechanism of inhibition of the HtrA proteases by specific Fabs are very briefly mentioned. The quality of the paper would be improved if the Authors discussed this in more detail, for example suggested in which diseases (other than GA) specific inhibition of a HtrA protease might be beneficial. The fact that many pathologies (eg. various cancers) correlate with a decrease in HtrA level/activity should also be taken into consideration.

7. There are some editorial errors:

Line 172 – is “inactivate”, should be “inactive”

Line 305: “HtrA1 is a highly conserved across species, including prokaryotes [10].” Please, correct this sentence.

Line 322: “We constructed the corresponding deletion mutants of HtrA2 and HtrA3 (HtrA2PD/PDZ-DLoopA and HtrA3PD/PDZ-DLoopA), to find out whether LoopA is also detrimental for other HtrA proteases.” Shouldn’t “detrimental” be exchanged with “essential” since the LA deletion mutants of the HtrAs are impaired?

Reviewer #2:

Remarks to the Author:

Summary

In their manuscript “Allosteric inhibition of HtrA1 activity by a “conformational lock” mechanism to treat age related macular degeneration” the authors Gerhardy et al. systematically study the mechanism by which a therapeutic Fab inhibits the catalytic activity of the human serine protease HTRA1. Overall, this work convincingly presents an allosteric mechanism by which Fab binding to a loop at significant distance from the active site stabilizes a conformation of key active center loops that is incompatible with substrate binding and catalytic activity. The experimental data comprises structural and functional assays that, altogether, support the conclusions well. HTRA1 is involved in a number of diseases, in particular age-related macular degeneration and neurodegenerative disorders. Therefore, strategies to modulate HTRA1 function and activity are of great therapeutic interest. With the study at hand, the authors elucidate a mechanism by which HTRA1 can be inhibited via Fab binding, providing a molecular explanation of specificity, which has been one of the key challenges in targeting trypsin-like serine proteases so far. Altogether I recommend publication of the work presented here with minor suggestions for improvements outlined below.

Key Aspects

The reported data identifies the epitope of Fab binding to HTRA1 to be a regulatory loop of the protease domain, LoopA. The binding interface is determined at high resolution using X-ray crystallography and confirmed by binding assays with mutant peptides. The heart of the study that contains most novelty and significance is based on the structural analysis of the HTRA1 protease domain in complex with the Fab using cryo-electron microscopy (cryo-EM). In the cryo-EM reconstruction, contacts between the Fab and HTRA1 are restricted to the LoopA confirming, together with supporting data, that LoopA binding is the major or even sole contributor to HTRA1 inhibition by the Fab. Strikingly, the active centers of the protease are consistently found to be in an inactive conformation. A thorough comparison with and discussion of known structural information on HTRA1 concludes that substrate binding and catalysis are inhibited by displacing several key residues and

structural elements away from a catalytically competent spatial configuration.

The work expands our knowledge of the mechanisms that regulate HTRA proteases, since the respective loops were found to have a regulatory impact on the activity of other HTRA proteases and have, to my knowledge, not yet been explored as regulatory determinants in detail. The work presented here therefore provides an important starting point for future studies aiming at elucidating the regulation and functional diversity of HTRA proteases in various biological contexts. The presented mechanism and approach to target HTRA proteases using the elucidated regulatory mechanism represents novelty beyond the research topic of HTRAs, since related serine proteases play crucial role in many biological and disease scenarios. Therefore, the results and conclusions will be of interest to a greater audience, justifying publication in a Nature Portfolio journal. The study combines a wealth of data to provide a mechanistic explanation of HTRA1 inhibition by Fab15H6.v4 as a therapeutic approach in macular degeneration and uncovers a novel contribution of the surface accessible loopA to the active center configuration.

The literature and present knowledge in the field are referred to appropriately and taken into account in a reasonable and balanced way. Furthermore, the manuscript is well written and incorporates the experimental data well into the argumentation, providing sufficient support for their conclusions, with minor limitations as discussed below.

The use of experimental approaches is justified and follows very clear and logical argumentation, including all critical controls. Cryo-EM data collection and processing were performed according to the quality standards in the field, all relevant methodological information, metrics and strategies are incorporated into the manuscript and supplementary data. The quality of the cryo-EM map, in particular in the most relevant part comprising the catalytic center of HTRA1, justifies the conclusions drawn based on the structural data.

The authors rightly state that the local resolution being highest in the center of the complex, comprising the active centers of the protease protomers supports the absence of a large extent of conformational variability. It would be interesting to see whether, in the course of the data processing pipeline, other conformations were discarded – for example doing 3D classification. Unfortunately, the authors do not provide detailed information or images on the classes that were obtained during 3D classification. These details could be informative regarding the presence of conformational heterogeneity despite the Fab being bound to the HTRA1 trimer. Naturally, the classes yielding the highest achievable resolution are typically selection for further processing or model building. However, the less well resolved classes may contain conformations that do have biological relevance but could be less well populated, more flexible, or less stable, leading to limited resolution values. I would like to encourage that the authors elaborate on this aspect, but I do not consider this a major issue, since it is most likely that the conformation discussed in the manuscript is indeed the one stabilized by the Fab and, thus, most relevant for the conclusions drawn.

Another minor point is that the stoichiometry of Fab to HTRA1 binding is not too well supported by experimental data. 3D classification could reveal the distribution of HTRA1 trimers bound to different numbers of Fab molecules. SEC is not sufficient to state the stoichiometry by itself. Furthermore, one of the reasons for a reduced local resolution of the Fab parts of the C1 reconstruction shown in S3e may be incomplete occupancy by the Fab. Therefore, the stoichiometry may not be strictly as claimed by the authors. It is not clear to me whether stating this potential variability would represent a limitation of the conclusions overall, so I do not consider this a major point. However, I feel that considering this limitation would be appropriate.

Regarding the MD simulations presented in Figs 3 and S6, it is interesting to see that the adoption of the catalytically incompetent conformation is energetically possible. It does look like, e.g. in Fig. 3b, that some conformations, as measured by the movement of L345 in that case, are energetically favored since Trial1 in Fig. 3b seems to get trapped in that conformation. It would be interesting to see whether, vice versa, the catalytically competent conformation(s) can be sampled when starting from the inactive conformation in the absence of the Fab.

A more fundamental limitation that I am not sure whether or not it can be addressed by the authors concerns the molecular details of the allosteric coupling of LoopA and the active center. The structural and biochemical observations are all very well founded and extensively supported by experimental data. As the authors correctly point out, however, the distance of the Fab binding site and the conformational rearrangements (or lack thereof) is striking. Still, very little information is provided//available as to how the allosteric signal (i.e., binding/stabilization of LoopA) is transmitted to the active center. Can the authors provide any more detailed reasoning on how this transmission could occur? How does LoopA compare in the inactive and active conformations reported in previous studies? The authors speculate that the movement and conformational flexibility of loopA may be required for allowing the active center loops to adopt catalytically competent configurations. However, introducing flexibility through GS linkers between LoopA and the adjacent beta-strand disrupts catalytic activity, suggesting a distinct and defined conformational coupling of LoopA and the active center loops. It would be important in my view to state whether the available structural data suggest any concrete molecular details of this coupling, e.g. specific amino acid residues or contacts that likely contribute to this mechanism. Should this not be possible, then this should be clearly and openly stated as an open question that needs further work in the future. It should be noted that I do not consider this an issue that would preclude publication of the data and conclusions as presented here. But I would like to ask for a more open/thorough discussion of this issue and the author should point this problem out as clearly as possible.

Along a similar line – it is nicely shown that replacing the HTRA1 LoopA with the respective loops from other HTRA family members reduces the catalytic activity of HTRA1, suggesting that the regulatory role of LoopA has diversified along with the emergence of different HTRA homologs. This is in itself a very interesting and highly relevant biological finding. However, in light of these findings, one has to be rather careful in drawing conclusions from the LoopA swapping experiments in which LoopA of other HTRAs is replaced with that of HTRA1, since it would be expected that the HTRA1 LoopA is not an adequate functional replacement of the native LoopA of other HTRA proteases. Therefore, conclusions based on Fab binding to an HTRA1 LoopA transferred to other HTRA1 are to be taken with a pinch of salt at least. Yes, it does suggest, in combination with other experiments, that the regulatory relevance of LoopA is indeed conserved, but beyond that, conclusions have to be stated very carefully, for example regarding the speculation that a similar inhibitory mechanism may also work for other HTRAs. It is an interesting and tempting idea, and the data points into that direction, but one has to be very careful here.

Could the authors speculate on the natural role that LoopA may play in HTRA proteases? Would it be a candidate regulatory element that may affect substrate specificity or local activity regulation within the cell? Is there any data in that direction yet, or any possibility exploring this idea? Given the different substrate specificities of HTRAs, and in particular the rather broad specificity of HTRA1, it would be very interesting to explore the differential effect that a manipulation of LoopA would have on the substrate selection by HTRA1 or other HTRA proteases.

Other Minor Points

- I would suggest to use capitalized HTRA1 since this is the convention for human proteins
- l. 172 reported resolution for the HTRA1 PD/SA is 3.6 Å in the Fig legend (l. 1008)
- l. 240 conformations
- l. 310 "to than HTRA1" – this sentence need some work

Reviewer #3:

Remarks to the Author:

The paper of Gerhardy et al represents a thorough characterization of the powerful antibody

Fab15H6.v4 that inhibits the serine protease HTRA1. This antibody is used in clinical trials aiming at treating geographic atrophy for which no cure is currently available. Therefore, this study is of general interest. The authors show that Fab15H6.v4 binds with very high affinity to the non-conserved loop LA of HTRA1, which is expected to solve issues of selectivity. A series of high-quality structural, biochemical, mutational and loop swapping experiments reveal that targeting loop LA can introduce conformational changes at the active site rendering the protease inactive.

There is one point I'd like to encourage the authors to consider when revising their paper. The point to consider is to distinguish between the effects of mutations or a ligand, in this case their antibody, that inhibit HTRA1 "indirectly" by targeting a site that is of considerable distance to the active site, from a true allosteric effect. The term allostery describes the communication between two binding sites, one of which, as in the case under investigation, can be the active site of an enzyme. Therefore, if the authors wish to claim that loop LA is an allosteric site, they should address the communication between loop LA and the active site e.g. by showing that targeting the active site by mutations or ligands interferes or improves binding of the antibody to loop LA. They have tools at hand to test this i.e. the activity based probes.

One problem for the case they are presenting is that loop LA doesn't interact with any structural elements of HTRA1 in the apo or active conformations, so, any direct effects on the elements of the activation domain etc are implausible. Note that a plethora of mutations are known to have far-reaching structural effects causing distortions of the active site, which doesn't mean that the underlying mechanism is allostery.

A minor but related issue is that the authors state that the antibody would not cause activation by inhibition. Yet, allostery would predict exactly such an effect. Perhaps I missed to see titrations of the antibody in HTRA1 activity assays including substoichiometric concentrations of the antibody.

REVIEWER COMMENTS

Reviewer #1 (Remarks to the Author):

The manuscript titled : “Allosteric inhibition of HtrA1 activity by a "conformational lock" mechanism to treat age related macular degeneration” by Stefan Gerhardy et al is a detailed, comprehensive study of regulation (inhibition) of the HtrA1 protease by the Fab15H6.v4 fragment of the anti-HtrA1 antibody, binding to the HtrA1 protease domain (PD).

The study is justified by the fact that HtrA1 is known to be associated with age related macular degeneration (AMD); AMD can be categorized into neovascular AMD and geographic atrophy (GA) and there is no treatment available for GA. A slightly modified version of the Fab15H6.v4, namely the Fab15H6.v4.D226, is currently being investigated as a novel treatment for GA in a Phase II clinical trial. Since the molecular basis of the HtrA1 activity inhibition by the Fab15H6.v4 has not been investigated in detail, this study is an important step towards understanding the mechanism of action of the clinically used Fab.

The Authors performed a series of experiments including cryogenic electron microscopy (cryoEM), X-ray crystallography, molecular dynamics simulations and biochemical assays and showed that the Fab binds to a small epitope contained in the LA loop of HtrA1 PD, at a comparatively long distance (about 30 Å) from the serine protease active center. They solved the structure of the HtrA1-Fab complex at a high resolution, identifying the amino acid residues involved in binding on both sides. They proved that the LA loop is important for activity regulation and showed that the Fab binding locks the HtrA1 active site (including the loops L1, L2, LD and residues H220, L345) in an inactive conformation - the active center of the Fab-bound HtrA1 is unable to bind a substrate and cleave it. The Fab binding was specific to the HtrA1 LA loop as it did not interact with chimeric HtrA1 carrying the LA loops of HtrA2-4. They also proved that Loop A regulates other HtrA human proteases (HtrA2/3), furthermore transfer of the HtrA1 LoopA to HtrA2 and HtrA3 enabled the Fab to inhibit the resulting chimeric proteases.

The Authors concluded that the long range allosteric mechanism by which Fab15H6.v4 inhibits the HtrA2 and HtrA3 chimeric proteases is similar to the conformational lock mechanism described for HtrA1. They also postulated that allosteric modulation of protease activity by a LoopA-specific antibody should be achievable for other members of the HtrA family. I believe this is probable and such approach could be used in future, should inactivation of any of the HtrAs be beneficial from the medical point of view. In my opinion, the experimental work has been done exceptionally well, the techniques are state of the art, each conclusion has been based on two or more experimental approaches, all conclusions are well based on the experimental data. The findings are novel and important for better understanding the regulation of human HtrA proteases, and are significant from the medical point of view. The results are well and clearly illustrated, the paper is very well written.

However, I have some comments.

1. The placing of control gels showing the proteins used in the experiments demonstrating substrate cleavage assays in the Supplementary Figure 10 may have saved some space but it is a most unusual practice. It is hard to believe that the protein gels run separately as controls are identical with the experimental gels shown in the figures to which the S10 refers. It is very inconvenient to find an

appropriate control in a different figure, especially in a situation when there is no information that such control exists. The controls should be shown together with the cleavage assays in Figs 1b, 5de, 6e. Besides, the legend to Fig. S10 is confusing: (1) it refers to Fig.5g – there is no Fig5g; (2) it refers to Fig. 7 – there is no Fig. 7; (3) the gel shown in S10d does not correspond to Fig5f, as stated; (4) one of the lanes in S10e is not described.

>> Agreed. We have transferred the control gels to the main figures for easier understanding and comparison. We have updated Figure 1b, Figure 5d,e,g and Figure 6e,f and the figure legend accordingly. We have also updated the supplementary Figure S11 and referenced all shown gels in the Source Data. All changes made in the text in response to the 3 reviewers are highlighted in yellow.

2. It would help to better understand the results if the molecular weights of the HtrA1PD, HtrA1PD-PDZ and HtrA1FL were stated in the text – the first chapter of Results would be a good place.

>> We have added the molecular weights to the results section (page 5), the figure 1 legend and the materials and methods section for improved clarity.

3. The cleavage of the HtrA1 specific substrates (i.e. DKK3, decorin, biglycan) is highly inefficient compared to the casein substrate processing (considering enzyme/substrate ratio and incubation time). I think the Authors should comment on this.

>> The reviewer is correct in observing that in the mentioned in vitro cleavage assays HtrA1 appears rather inefficient compared to the generic casein-BODIPY assay. We now added time- and concentration dependent kinetic studies to the Supplementary Section to provide quantitative data regarding the cleavage efficiency of HTRA1^{FL} (Supplementary Figure S1a-d). Full length HTRA1 at a concentration of 1 μ M completely cleaved DKK3 after 8-16 hours incubation at 37°C (Supplementary Figure S1a), in agreement with a previous study (Tom I. et al. 2020 PNAS). Moreover, quantification of substrate cleavage in two independent experiments showed that BIGLYCAN is very efficiently cleaved by HTRA1 (50% cleaved at 0.01-0.1 μ M HTRA1 during 260 min incubation; Supplementary Figure S1b). Cleavage of both DKK3 and DECORIN were less efficient, requiring longer incubation (16 h) and higher HTRA1 concentrations: 50% cleavage of DKK3 and DECORIN was achieved with 0.01-0.1 μ M and with 0.1-0.5 μ M HTRA1, respectively (Supplementary Figure S1c, d). The results indicate that the cleavage efficiency of HTRA1 depends on the type of substrate (BIGLYCAN >> DKK3 > DECORIN). We chose a 1 μ M HTRA1 concentration for our substrate cleavage assays (Figure 1) in order to show that Fab15H6.v4 is able to fully inhibit HTRA1 under quite stringent conditions (i.e. at high enzyme concentration and over an extended period of time). This is now mentioned in the revised results section (page 5), including a reference to the substrate cleavage kinetics shown in Supplementary Figure 1.

4. The Authors showed that the full length HtrA1 and HtrA1-PD was inhibited by the Fab, and that the chimeric HtrA2-PD and HtrA3-PD carrying the LA loops of HtrA1 were also inhibited by the Fab. They postulated that the HtrA2/HtrA3 could be inhibited by specific Fabs interacting with their LA loops. However, the naturally existing HtrA2/3 have PDZ domains which are positioned differently with respect to PD (including LA loops) compared to HtrA1 - this is especially visible in HtrA2. Did the Authors try to model the Fab- HtrA2/3 interactions to see if there is no steric hindrance due to the different position of PDZ? They should comment on this in the paper.

>> We have generated an additional Supplementary Figure (S10) using our solved HTRA1:Fab structure, which was superimposed with the available crystal structures of HTRA2 (pdb 1LCY) and HTRA3 (4RI0) to identify potential steric interference with the PDZ domain. In the HTRA3 structure (pdb 4RI0), there

is a clash between the PDZ domain and the Fab. However, this clash could be easily remedied by a small movement of the flexible linker region (connecting the PDZ and the catalytic domain), which would reposition the PDZ away from the bound Fab. The idea of a flexible linker is supported by the fact that the PDZ domain is only resolved in one of the three HTRA3 protomers in pdb 4RI0.

Regarding HTRA2, the PDZ domain is located next to the Fab and would probably not interfere with Fab binding in the chimeric proteins. Additionally, HTRA2 also must have a flexible linker to reposition its PDZ domain, since the structure (pdb 1LCY) shows the PDZ domain occupying the active site region, which would render the enzyme inactive. In further support of PDZ domain mobility for all HTRA family members, the structure of HtrA1^{PD-PDZ} determined by Truebestein et al. 2011 (Nat. Struct. Mol. Biol. 18:386) only showed the protease domain trimer, while the PDZ domain was not defined by electron density. Therefore, since HTRA1-3 structures indicate PDZ mobility, the Fab binding to the chimeric proteases may occur without major obstructions. This conclusion is further supported by our biochemical data:

1. The HTRA2 and HTRA3 constructs we used in enzymatic and in binding assays contained the PDZ domains (HTRA2^{PD/PDZ} and HTRA3^{PD/PDZ}) (see material and methods). We could show that the Loop A chimeric proteins, which retain their PDZ domain, supported high affinity for Fab binding (Kd in picomolar range), consistent with unobstructed Fab access to the LoopA.
2. Similarly, binding of Fab15H6.v4 to the HTRA2 and HTRA3 chimeric proteins completely inhibited enzymatic activity and reduced labeling by the activity-based probe TAMRA-ABP. Therefore, we conclude that the PDZ domain does not interfere with Fab binding in solution.

We now explicitly state that the HTRA2 and HTRA3 chimeric constructs contain the PDZ domain (Figure 6 legend) and have added a supplementary figure (Supplementary Figure S10) which is discussed in the updated results section (page 11)

5. Methods: Please, state the time range in which the cleavage of the Bodipy-casein was recorded.

>> This is now corrected. The section in the material and methods part now reads:

“Cleavage of BODIPY-casein was measured at 37°C during a 10 min time period using a SPECTRAmax M5 microplate reader (Molecular Devices) and the linear rates of substrate cleavage were determined.”
(page 20)

6. In the Discussion, the structural and kinetic data concerning the HtrA1 inhibition are discussed in-depth. Contrary to this, the possible medical applications of the proposed general mechanism of inhibition of the HtrA proteases by specific Fabs are very briefly mentioned. The quality of the paper would be improved if the Authors discussed this in more detail, for example suggested in which diseases (other than GA) specific inhibition of a HtrA protease might be beneficial. The fact that many pathologies (eg. various cancers) correlate with a decrease in HtrA level/activity should also be taken into consideration.

> Thank you for pointing this out. We have extended the final paragraph in the discussion highlighting a potential application of Fab15H6.v4 for the treatment of arthritic disease and discuss other HTRA1-associated diseases.

The last paragraph of the discussion now reads (page 13):

“In conclusion, our study provides the first detailed insight into the mode of inhibition of the trimeric serine protease HTRA1 by a clinical Fab fragment. Together with the reported safety, tolerability, and pharmacodynamics of Fab15H6.v4.D221 [31-33], our findings provide an improved

understanding of this Fab-based therapeutic approach to regulate HTRA1 enzymatic activity in GA. The therapeutic potential of Fab15H6.v4.D221 may not be limited to AMD, as HTRA1 has been implicated in other chronic diseases, such as osteoarthritis [54, 55]. HTRA1 expression is increased in synovial fluid of osteoarthritis patients and may contribute to cartilage degradation [18, 56]. Specific inhibition of HTRA1 activity could protect the integrity of the pericellular matrix [54] and may serve as a novel therapeutic opportunity for treatment of arthritic diseases. In contrast, the role of HTRA1 in CARASIL [13, 21, 22], certain cancers [57, 58] and in Alzheimer's disease [19, 59] may require different therapeutic approaches aimed at enhancing, rather than inhibiting HTRA1 activity.

7. There are some editorial errors:

Line 172 – is “inactivate”, should be “inactive”

>> Corrected.

Line 305: “HtrA1 is a highly conserved across species, including prokaryotes [10].” Please, correct this sentence.

>> We have made the correction and rewritten this small paragraph to highlight the possibility to use our Fab in preclinical disease models (page 9):

“The HTRA1 protease domain including the LoopA epitope is highly conserved across mammalian species (Supplementary Figure S8a, [10]). Thus, our structural model predicts that Fab15H6.v4 could also be used in different disease model organisms. Indeed, Fab15H6.v4 inhibits murine HTRA1 (Supplementary Figure S8b) and blocks the labeling of its active site by TAMRA-ABP (Supplementary Figure S8c) similar to human HTRA1 (Figure 1).”

Line 322: “We constructed the corresponding deletion mutants of HtrA2 and HtrA3 (HtrA2PD/PDZ-DLoopA and HtrA3PD/PDZ-DLoopA), to find out whether LoopA is also detrimental for other HtrA proteases.” Shouldn’t “detrimental” be exchanged with “essential” since the LA deletion mutants of the HtrAs are impaired?

>> Thank you for this correction. We have made the change as suggested (page 10).

Reviewer #2 (Remarks to the Author):

Summary

In their manuscript “Allosteric inhibition of HtrA1 activity by a “conformational lock” mechanism to treat age related macular degeneration” the authors Gerhardy et al. systematically study the mechanism by which a therapeutic Fab inhibits the catalytic activity of the human serine protease HTRA1. Overall, this work convincingly presents an allosteric mechanism by which Fab binding to a loop at significant distance from the active site stabilizes a conformation of key active center loops that is incompatible with substrate binding and catalytic activity. The experimental data comprises structural and functional assays that, altogether, support the conclusions well. HTRA1 is involved in a number of diseases, in particular age-related macular degeneration and neurodegenerative disorders. Therefore, strategies to modulate HTRA1 function and activity are of great therapeutic interest. With the study at hand, the authors elucidate a mechanism by which HTRA1 can be inhibited via Fab binding, providing a molecular explanation of specificity, which has been one of the key challenges in targeting

trypsin-like serine proteases so far. Altogether I recommend publication of the work presented here with minor suggestions for improvements outlined below.

Key Aspects

The reported data identifies the epitope of Fab binding to HTRA1 to be a regulatory loop of the protease domain, LoopA. The binding interface is determined at high resolution using X-ray crystallography and confirmed by binding assays with mutant peptides. The heart of the study that contains most novelty and significance is based on the structural analysis of the HTRA1 protease domain in complex with the Fab using cryo-electron microscopy (cryo-EM). In the cryo-EM reconstruction, contacts between the Fab and HTRA1 are restricted to the LoopA confirming, together with supporting data, that LoopA binding is the major or even sole contributor to HTRA1 inhibition by the Fab. Strikingly, the active centers of the protease are consistently found to be in an inactive conformation. A thorough comparison with and discussion of known structural information on HTRA1 concludes that substrate binding and catalysis are inhibited by displacing several key residues and structural elements away from a catalytically competent spatial configuration.

The work expands our knowledge of the mechanisms that regulate HTRA proteases, since the respective loops were found to have a regulatory impact on the activity of other HTRA proteases and have, to my knowledge, not yet been explored as regulatory determinants in detail. The work presented here therefore provides an important starting point for future studies aiming at elucidating the regulation and functional diversity of HTRA proteases in various biological contexts. The presented mechanism and approach to target HTRA proteases using the elucidated regulatory mechanism represents novelty beyond the research topic of HTRAs, since related serine proteases play crucial role in many biological and disease scenarios. Therefore, the results and conclusions will be of interest to a greater audience, justifying publication in a Nature Portfolio journal. The study combines a wealth of data to provide a mechanistic explanation of HTRA1 inhibition by Fab15H6.v4 as a therapeutic approach in macular degeneration and uncovers a novel contribution of the surface accessible loopA to the active center configuration.

The literature and present knowledge in the field are referred to appropriately and taken into account in a reasonable and balanced way. Furthermore, the manuscript is well written and incorporates the experimental data well into the argumentation, providing sufficient support for their conclusions, with minor limitations as discussed below.

The use of experimental approaches is justified and follows very clear and logical argumentation, including all critical controls. Cryo-EM data collection and processing were performed according to the quality standards in the field, all relevant methodological information, metrics and strategies are incorporated into the manuscript and supplementary data. The quality of the cryo-EM map, in particular in the most relevant part comprising the catalytic center of HTRA1, justifies the conclusions drawn based on the structural data.

The authors rightly state that the local resolution being highest in the center of the complex, comprising the active centers of the protease protomers supports the absence of a large extent of conformational variability.

It would be interesting to see whether, in the course of the data processing pipeline, other conformations were discarded – for example doing 3D classification. Unfortunately, the authors do not

provide detailed information or images on the classes that were obtained during 3D classification. These details could be informative regarding the presence of conformational heterogeneity despite the Fab being bound to the HTRA1 trimer. Naturally, the classes yielding the highest achievable resolution are typically selection for further processing or model building. However, the less well resolved classes may contain conformations that do have biological relevance but could be less well populated, more flexible, or less stable, leading to limited resolution values. I would like to encourage that the authors elaborate on this aspect, but I do not consider this a major issue, since it is most likely that the conformation discussed in the manuscript is indeed the one stabilized by the Fab and, thus, most relevant for the conclusions drawn.

>> The referee is correct that some conformations of HTRA1 may occur in our Fab-bound samples that are distinct from that resolved in our deposited maps. Following this suggestion, we re-examined for the HTRA1^{PD/SA} mutant dataset the five 3D classes obtained before particle polishing against a single map (refer to Fig S2). The five classes contained approximately equal numbers of particle images (19.3%, 20.6%, 18.5%, 25.0%, 16.6%) and reached similar resolutions (4.0 Å, 3.9 Å, 4.25Å, 4.0 Å, 4.1 Å); the main visible differences between the classes was that the Fabs were "splayed out" at slightly different angles to the symmetry axis (with variations on the order of 5 degrees); visual inspection of each of the maps revealed no obvious conformational changes near the active site, nor in LoopA, though mainchain connectivity was not continuous in all 5 maps, and not all side chains were equally well resolved in the five maps, as is typical of maps at this resolution without fully-refined alignment parameters (as a reminder, the particles at this stage had not been "polished" by RELION).

Specifically, while the His220 sidechain was resolved in all 5 maps in its non-competent orientation, the Asp250 side chain was not, and its rotameric orientation was more ambiguous; further, the loop formed by Glu247-Asp250 (in loop C) was poorly resolved in some of the maps, suggesting this loop may remain somewhat dynamic even when HTRA1 is Fab-bound. However, this loop is not part of the described conformational lock. Other regions surrounding the catalytic site and especially the active center Loops affected by Fab binding (Loop1, Loop2 and LoopD) appeared to be well conserved between all five classes.

We have expanded the Supplementary Figure Legend S2 to further address this point:

"Figure S2 a. Peak fractions of the HTRA1^{PD/SA}:Fab15H6.v4 complex from Figure 1d were reduced and loaded onto 12% NuPAGE Gel (Thermo Fisher Scientific), run in 1x MES SDS Running buffer (Thermo Fisher Scientific) and stained with SimplyBlue SafeStain. The HTRA1^{PD/SA} and Fab co-migrate under reducing conditions (~25kDa). b. Representative micrograph of HTRA1^{PD/SA}:Fab15H6.v4 complex. c. 2D class averages of HTRA1^{PD/SA}:Fab15H6.v4 complex. d. Cryo-EM processing workflow using cisTEM. The obtained five 3D classes showed different angles of Fab15H6.v4 relative to the HTRA1 trimer. Visual inspection of each of the map showed similar conformational distortion in the active site. Further refined by RELION particle polishing using the best of the five maps enabled resolving cryo-EM maps with increased resolution.

(All changes made in the text in response to the 3 reviewers are highlighted in yellow.)

Another minor point is that the stoichiometry of Fab to HTRA1 binding is not too well supported by experimental data. 3D classification could reveal the distribution of HTRA1 trimers bound to different numbers of Fab molecules. SEC is not sufficient to state the stoichiometry by itself. Furthermore, one of the reasons for a reduced local resolution of the Fab parts of the C1 reconstruction shown in S3e may be incomplete occupancy by the Fab. Therefore, the stoichiometry may not be strictly as claimed by the authors. It is not clear to me whether stating this potential variability would represent a limitation of the conclusions overall, so I do not consider this a major point. However, I feel that considering this limitation would be appropriate.

>> This is a valid point, and we evaluated the cryo-EM data very carefully and carried out an enzyme titration experiment to more firmly establish the stoichiometry. Collectively, our data strongly support a 1:3 stoichiometry (HTRA1 trimer:Fab).

1. Cryo-EM: We cannot prove the absence of sub-stoichiometric complexes in our sample. However, several lines of evidence from our image processing results suggest that 1:3 stoichiometry is indeed overwhelmingly dominant: First, we attempted 3D classification of our datasets in C1 multiple times, including with different software packages, and could not obtain even minority classes with one or more missing Fabs. Secondly, we attempted the same after symmetry expansion and masking so that only one monomer:Fab part of the trimeric assembly was considered, and again failed to obtain a 3D class without bound Fab. Thirdly, we never observed any 2D class averages with only 2 (or 1) Fab bound. As the referee will be aware, none of these observations completely discards the possibility that some complexes exist in sub-stoichiometric states. However, when taken together with the very high affinities of this Fab for Htra1 protein and even for the synthetic loop A peptide measured by SPR, in our opinion these lines of evidence are strongly suggestive of overwhelmingly 1:3 stoichiometric HTRA1 trimer:Fab binding.
2. Size exclusion chromatography: SEC experiments gave the first indication that 3 Fabs bound to each HTRA1 trimer. This conclusion was based on the mass of the Fab:HTRA1^{PD/SA} peak eluted around 200kDa based on a standard curve with MW standards used in the same SEC system. This mass is approximating the theoretical mass of the predicted 1:3 (HTRA1 trimer:Fab) complex of 228kDa. We also formed the complex of Fab and HTRA1^{FL/SA} and obtained similar results suggesting a 1:3 complex: The complex eluted at 300-400 kDa approximating the theoretical mass of 316 kDa of the predicted 1:3 complex (HTRA1 trimer:Fab).
3. Enzyme titration: To further address the stoichiometry of the complex, we have now performed enzyme titration experiments (see extended Supplementary Figure S1e below). The Fab concentration achieving 100% inhibition is deduced from the X-axis intercept in a linear plot depicting the enzyme activity as a function of Fab concentration. This value was determined to be 32.55 ± 0.79 nM Fab for 10nM HTRA1^{PD} trimer (n=3), which is very close to the theoretical value of 30 nM Fab of a 1:3 complex (HTRA1 trimer:Fab). We have added the titration results as Supplementary Figure S1e and describe it in the result section (page 5):

“Additionally, titration of HTRA1^{PD} with Fab15H6.v4 showed a progressive inhibition of HTRA1^{PD} enzyme activity at sub-stoichiometric concentration reaching complete inhibition at a 1:3 (HTRA1 trimer:Fab15H6.v4) molar ratio (Supplementary Figure S1e), consistent with the stoichiometry determined by size exclusion chromatography (Figure 1d).”

Supplementary Figure S1e. Titration of 10 nM HTRA1^{PD} trimer with sub-stoichiometric concentrations of Fab15H6.v4 (32 nM to 6.5 nM). Full inhibition is achieved at a concentration of 32.5 nM of Fab15H6.v4, indicating a 1:3 complex (HTRA1 trimer:Fab).

Regarding the MD simulations presented in Figs 3 and S6, it is interesting to see that the adoption of the catalytically incompetent conformation is energetically possible. It does look like, e.g. in Fig. 3b, that some conformations, as measured by the movement of L345 in that case, are energetically favored since Trial1 in Fig. 3b seems to get trapped in that conformation. It would be interesting to see whether, vice versa, the catalytically competent conformation(s) can be sampled when starting from the inactive conformation in the absence of the Fab.

>> Thank you for this suggestion. Accordingly, we ran additional simulations (3 trials) starting from an inactive conformation of apo-HTRA1 (3TJN_chainA) and focused on the same hallmark residues as reported.

When starting from the inactive conformation, the HTRA1 catalytic triad can indeed reach an active conformation indicated by the distance between H220 and S328 (distance His-Ser = 3.6 Å, green line; see above, left panel). Trial 3 (in teal) does sample the active conformation which would be able to support the activation of the catalytic serine S328. Trial 2 (in red) briefly visits this conformation, achieving H-bonding distance to the S328 at around 150ns. Trial 1 remained in the inactive conformation during the 1000 ns MD simulation.

In respect to the second hallmark residue (L345) we could not observe a stable occupancy of the active conformation (distance Leu-Ser = 8.9 Å, green line, see above right panel). When starting from the

inactive conformation, the data suggests that L345 stays in the non-competent conformation, occluding the S1 pocket.

At this point it remains unclear why the inactive conformation (3TJN_chainA) is unable to reach a fully formed active site conformation, which is clearly present in the crystallographic studies (Eigenbrot et al Structure 2012). It is possible that the time scale of the MDS experiments is too short and that the expected changes may only take place in much extended time frames. Since this question does not directly pertain to the reported allosteric inhibition mechanism, we did not carry out additional experiments and hope that this is agreeable to the reviewer.

A more fundamental limitation that I am not sure whether or not it can be addressed by the authors concerns the molecular details of the allosteric coupling of LoopA and the active center. The structural and biochemical observations are all very well founded and extensively supported by experimental data. As the authors correctly point out, however, the distance of the Fab binding site and the conformational rearrangements (or lack thereof) is striking. Still, very little information is provided//available as to how the allosteric signal (i.e., binding/stabilization of LoopA) is transmitted to the active center. Can the authors provide any more detailed reasoning on how this transmission could occur?

>> Thank you for this comment. Indeed, the molecular details of LoopA connectivity to the active site loops remains unclear at this moment. There are several different hypotheses about the transmission of the allosteric signals and diverse computational approaches to identify these communication networks have been reported (e.g. Bedem et al Nature Methods 2013, Feher et al Curr Opin Structural Biology 2017 Review). We have tried to address the communication between these two sites through a computational approach and by an independent mutagenesis experiment, in which we mutated residues in the LoopB, that is positioned between the LoopA and the active site. However, the mutations, including insertions and deletions did not disrupt the connectivity, as Fab15H6.v4 was still able to inhibit activity (data not shown). Moreover, we have utilized an ensemble correlation network analysis (Grant et al., 2021) to determine potential paths of communication between residues at LoopA (R197 and F194) and the catalytic center using the 1 μ s molecular dynamic simulations for the uninhibited apo-HTRA1 structure (3TJN_chainB) and the Fab-restrained structure. However, the results did neither identify a unique allosteric network from LoopA to the critical catalytic His220 residue, nor to the Loop2 residue Leu345 with any reasonable confidence. Thus, while the employed methodology did not yield any tangible results, it is possible that other computational approaches might be more successful, but this is beyond our expertise.

Therefore, the connectivity between the LoopA epitope and the active site conformations remains unresolved, although Fab binding to LoopA clearly induces detrimental conformational changes within the active site. We have now stated this fact more forcefully in the discussion section (page 13).

References:

Bedem H. et al. (2013). Automated identification of functional dynamic networks from X-ray crystallography, Nat Methods. 2013 September ; 10(9): 896–902. doi:10.1038/nmeth.2592

Feher VA et al. (2014) Computational approaches to mapping allosteric pathways. Curr Opin Struct Biol 25:98–103. doi: [10.1016/j.sbi.2014.02.004](https://doi.org/10.1016/j.sbi.2014.02.004)

Grant, B. J., Skjærven, L., and Yao, X. (2021). The Bio3D packages for structural bioinformatics. Protein Sci 30, 20–30. doi: [10.1002/pro.3923](https://doi.org/10.1002/pro.3923).

How does LoopA compare in the inactive and active conformations reported in previous studies?

>> Compared to other inactive (pdb 3TJN_chainA, 3TJO and 3NUM) and active structures (pdb 3NWU, 3TJN_chainB, 3NZI), Fab binding slightly reorients the LoopA in respect to the protease domain (see Supplementary Figure S4d). This is captured in the results section (page 6):

"In comparison to the crystal structures of unbound apo-HTRA1, the LoopA in both HTRA1:Fab15H6.v4 complexes is twisted outwards by almost 20° and the β-strand stem partially unfolds at V199-V201 into a loop structure (Supplementary Figure S4d)."

The authors speculate that the movement and conformational flexibility of loopA may be required for allowing the active center loops to adopt catalytically competent configurations. However, introducing flexibility through GS linkers between LoopA and the adjacent beta-strand disrupts catalytic activity, suggesting a distinct and defined conformational coupling of LoopA and the active center loops. It would be important in my view to state whether the available structural data suggest any concrete molecular details of this coupling, e.g. specific amino acid residues or contacts that likely contribute to this mechanism. Should this not be possible, then this should be clearly and openly stated as an open question that needs further work in the future. It should be noted that I do not consider this an issue that would preclude publication of the data and conclusions as presented here. But I would like to ask for a more open/thorough discussion of this issue and the author should point this problem out as clearly as possible.

>> Based on the GS linker experiments we suspected that the stem region (two antiparallel beta strands connected to Loop A) is an important element in the allosteric network. However, structural analysis did not present any convincing clues on how this stem region is connected to the active site loops on the single residue level. We speculated that LoopA communicated via the stem region directly to the active center either through main chain and/or side chain VDW contacts, or through a hydrogen bonding network on the surface of the protease domain. Moreover, as mentioned above we have utilized an ensemble correlation network analysis (Grant et al., 2021) to determine paths of communication between LoopA and the active site, but we were unsuccessful. Therefore, we are lacking any scientific evidence to support a specific pathway hypothesis. Following the reviewer's comment we have now stated this more explicitly in discussion section (page 13).

"However, it remains currently unclear how the allosteric signal is transmitted from the distant LoopA to the catalytic center."

Along a similar line – it is nicely shown that replacing the HTRA1 LoopA with the respective loops from other HTRA family members reduces the catalytic activity of HTRA1, suggesting that the regulatory role of LoopA has diversified along with the emergence of different HTRA homologs. This is in itself a very interesting and highly relevant biological finding. However, in light of these findings, one has to be rather careful in drawing conclusions from the LoopA swapping experiments in which LoopA of other HTRAs is replaced with that of HTRA1, since it would be expected that the HTRA1 LoopA is not an adequate functional replacement of the native LoopA of other HTRA proteases. Therefore, conclusions based on Fab binding to an HTRA1 LoopA transferred to other HTRA1 are to be taken with a pinch of salt at least. Yes, it does suggest, in combination with other experiments, that the regulatory relevance of LoopA is indeed conserved, but beyond that, conclusions have to be stated very carefully, for example regarding the speculation that a similar inhibitory mechanism may also work for other HTRAs. It is an interesting and tempting idea, and the data points into that direction, but one has to be very careful here.

>> We agree that in the loopA swap experiments, the HTRA1-LoopA might not serve as an exact functional replacement for the wildtype LoopA sequences in HtrA2 and HtrA3. Accordingly, we have made changes in the text which address this issue (see below) and hope this satisfies the reviewer's objections. Nevertheless, despite lacking the ultimate experimental proof, we believe that the LoopA-chimeras are reliable substitutes for reflecting the regulatory role of LoopA within the HTRA family for

the following reasons. As shown in Figure 6a, the engraftment of HTRA1-LoopA into HtrA2 and HtrA3 did not diminish the catalytic activity of the chimeras compared to their wildtype versions, suggesting a true functional replacement in terms of catalytic activity. Therefore, we concluded that these chimeras represent a valid system to examine the presence of an allosteric inhibition mechanism in HTRA2 and HTRA3 using Fab15H6.v4 as a surrogate for an antibody that would bind to the native LoopA of HTRA2 and HTRA3, since we did not have any such antibodies available (this is mentioned in the results section page 11).

We attempted to engineer Fab15H6.v4 into an HTRA2-binding antibody by modifying CDR loops based on a structural model of HTRA2-loopA bound to Fab15H6.v4. We identified residue D52 located on V_HCDR2 that seemed important for the specificity towards HtrA1. Therefore, we generated the Fab15H6.v4 mutant V_HCDR2-D52A in the hope to eliminate the steric clash with HTRA2-LoopA. The Fab15H6.v4 D52A mutant showed significantly decreased affinity towards HTRA1 (459 nM vs. 118 pM wildtype HTRA1) but was still unable to recognize HTRA2 (and HTRA3) as we did not detect any binding by SPR. We made the following change in the discussion section (page 13):

“Like HTRA1, the other HTRA family members exhibit conformational plasticity within the catalytic center [51-53], and might be susceptible to a LoopA-directed allosteric lock mechanism akin to HTRA1. This hypothesis was tested by use of chimeric HTRA2 and HTRA3 proteases containing the engrafted LoopA from HTRA1, which provided high affinity Fab binding to the chimeras. We observed complete enzyme inhibition and reduced TAMRA-ABP labeling of these chimeric proteins by Fab15H6.v4, suggesting a factual transfer of the allosteric inhibition mechanism and confirming the hypothesis. A caveat concerning this conclusion is that it is based on the assumption that the transferred HTRA1-LoopA served as a true functional replacement of the native LoopA sequences of HTRA2 and HTRA3. Nevertheless, the results raise the intriguing possibility that antibodies selected for binding to the natural LoopA of HTRA2-4 proteases could effectively inhibit their enzyme activities by the same conformational lock mechanism as observed for Fab15H6.v4 and HTRA1.”

Could the authors speculate on the natural role that LoopA may play in HTRA proteases? Would it be a candidate regulatory element that may affect substrate specificity or local activity regulation within the cell? Is there any data in that direction yet, or any possibility exploring this idea? Given the different substrate specificities of HTRAs, and in particular the rather broad specificity of HTRA1, it would be very interesting to explore the differential effect that a manipulation of LoopA would have on the substrate selection by HTRA1 or other HTRA proteases.

>> These are interesting questions and we agree that they are worth exploring. In fact, we have started an effort to address the role of LoopA in terms of local activity regulation, but this work is still at a very early stage. Traditionally, the Loop A in trypsin fold serine protease can participate (not in every case) in substrate and natural inhibitor recognition. The only natural inhibitor reported for HTRA1 is the serpin α 1-antitrypsin (Hu et al. JBC 273:34406, 1998). In some protease:serpin complexes the LoopA is engaged in serpin contacts and it is possible that this may be the case for α 1-antitrypsin interaction with HTRA1, but we have not examined this.

Beyond α 1-antitrypsin, there are no other known natural inhibitors and, to our knowledge, no systematic approaches to identify natural inhibitors have been carried out. This seems particularly pertinent considering that HTRA1, 3 and 4 don't follow the classic zymogen activation paradigm of trypsin fold serine proteases, as they are produced as fully active enzymes and do not require activation cleavage. In light of this, the LoopA seems to be an attractive structural element that could

be the target of a natural regulator/inhibitor. However, we have not pursued this hypothesis, as we were more interested in elucidating the inhibitory mechanism of Fab15H6.v4.

Another possibility we thought about, and which is also pointed out by the reviewer, is that LoopA could directly or indirectly through its extension (the beta strand "stem") participate in substrate recognition and substrate specificity, which would follow the canonical function of members of the

trypsin fold serine protease family. For instance, in an overlay of the apo-HTRA1 structure (pdb 3tjn) with the structure of thrombin with bound peptide substrate (PAR1 peptide; pdb 3LU9, see overlay below) we see that the substrate S6' residue (Arg in red) is positioned close to the LoopA beta strand stem of HTRA1. The P6' pocket does indeed show a slight preference for negatively charged or small amino acids (Tom I et al 2020, Figure 2a: <https://www.pnas.org/doi/epdf/10.1073/pnas.1917608117>). Thus, it is conceivable that the stem region of LoopA might contribute to substrate specificity of HTRA1.

We have added a sentence to the discussion section to speculate about the role of LoopA in activity regulation (discussion page 13):

" The importance of LoopA for HTRA catalytic activity raises the possibility that potential interactors of LoopA could regulate HTRA protease activity in the local environment. The regulation of HTRA protease activity is particularly pertinent considering that the secreted HTRA members (HTRA1, 3, 4) do not follow the zymogen activation paradigm of trypsin fold serine proteases but are secreted as trimeric active enzymes not requiring activation cleavage."

Other Minor Points

- I would suggest to use capitalized HTRA1 since this is the convention for human proteins
>> This is correct. We have updated the correct convention throughout the text and Figures.

- l. 172 reported resolution for the HTRA1 PD/SA is 3.6 Å in the Fig legend (l. 1008)
>> This error was corrected in the Figure 2 legend.

- l. 240 conformations

>> Corrected.

- l. 310 “to than HTRA1” – this sentence need some work

>> Corrected.

Reviewer #3 (Remarks to the Author):

The paper of Gerhardy et al represents a thorough characterization of the powerful antibody Fab15H6.v4 that inhibits the serine protease HTRA1. This antibody is used in clinical trials aiming at treating geographic atrophy for which no cure is currently available. Therefore, this study is of general interest. The authors show that Fab15H6.v4 binds with very high affinity to the non-conserved loop LA of HTRA1, which is expected to solve issues of selectivity. A series of high-quality structural, biochemical, mutational and loop swapping experiments reveal that targeting loop LA can introduce conformational changes at the active site rendering the protease inactive.

There is one point I'd like to encourage the authors to consider when revising their paper. The point to consider is to distinguish between the effects of mutations or a ligand, in this case their antibody, that inhibit HTRA1 "indirectly" by targeting a site that is of considerable distance to the active site, from a true allosteric effect. The term allostery describes the communication between two binding sites, one of which, as in the case under investigation, can be the active site of an enzyme.

Therefore, if the authors wish to claim that loop LA is an allosteric site, they should address the communication between loop LA and the active site e.g. by showing that targeting the active site by mutations or ligands interferes or improves binding of the antibody to loop LA. They have tools at hand to test this i.e. the activity based probes.

>> We thank the reviewer for the insightful comments and suggestions. We have used the term "allostery" as it is commonly applied to allosteric antibodies/peptides that inhibit catalysis of related trypsin fold serine proteases (Blouse 2009; Dennis 2000; Kromann-Hansen 2013; Koschubs 2012; Kromann-Hansen 2017; Schaefer 2019; Ganesan 2009). In all these examples the allosteric mechanism is supported by crystal structures showing that the allosteric antibody/nanobody/peptide binds outside the active site region, but influences the conformational state of activation domain loops. In each case this leads to reduced catalytic activity, providing evidence for the communication from the allosteric site to the active site, as we also show for Fab15H6.v4 in our study.

According to the above mentioned studies, the reverse communication, i.e. from the active site to the allosteric site, identified by structural studies, seems less effectual. For instance, the occupation of the uPA active site by a covalent substrate mimetic had relatively small effects on antibody/nanobody binding (1.5 to 3-fold binding loss; Blouse 2009; Kromann-Hansen 2017). This is in full agreement with our studies as shown in figure 1e. We pre-formed the covalent complex of HtrA1^{PD} with the substrate mimicking 7-mer peptide ABP and measured binding to Fab15H6.v4 by SPR. The determined K_D values were 118 ± 8 pM for HtrA1^{PD} and 207 ± 8 pM for the HtrA1^{PD}:7-mer peptide complex, indicating a Fab binding loss of less than 2-fold to the enzyme:7-mer peptide complex, in accordance with the antibody/nanobody binding results for uPA .

While this could be interpreted as being consistent with a bidirectional communication between the allosteric sites, the efficiency for the two directions is clearly not equivalent. This may be due to the strong binding affinities of the Fab/antibodies/nanobodies examined and the nature of their binding site, which are surface-exposed loops. The Fab15H6.v4 binds with very high affinity to both HtrA1 and to the synthetic loop A peptide (K_D 0.7nM; Supplementary Table 5), which is unstructured in solution (as indicated by NMR experiments; data now shown); therefore, it would be difficult to imagine that active site occupancy would effectively prevent Fab binding to the HtrA1-LoopA.

We have added the paragraph below to the results section to discuss our "reverse direction" results in the context of published studies related to allosteric antibodies of trypsin fold serine proteases (page 5-6):

"Furthermore, we investigated whether Fab15H6.v4 might bind to the active site itself. When the active site of HTRA1^{PD} was blocked with a heptameric activity-based probe (biotin-DPMFKLV-phosphonate derived from [24], 7-mer ABP), Fab15H6.v4 still bound with sub-nanomolar affinity (HTRA1^{PD} + 7-mer ABP: $K_D = 0.21$ nM, compared to the uninhibited HTRA1^{PD}: $K_D = 0.12$ nM, Figure 1e Supplementary Table 1), indicating that the Fab does not bind to or near the HTRA1 active site itself, but to an allosteric site. The relatively small decrease in affinity of Fab15H6.v4 to the active-site blocked HTRA1^{PD} is consistent with the reported 1.5 to 3-fold affinity reductions to active site-blocked urokinase-type plasminogen activator (uPA) by allosteric antibodies [37, 38]."

(All changes made in the text in response to the 3 reviewers are highlighted in yellow.)

References:

- Blouse G. et al JBC 2009: [https://www.jbc.org/article/S0021-9258\(20\)71067-7/fulltext](https://www.jbc.org/article/S0021-9258(20)71067-7/fulltext)
Dennis M. et al Nature 2000: <https://www.nature.com/articles/35006574>
Kromann-Hansen T. et al. Biochemistry 2013: <https://pubs.acs.org/doi/10.1021/bi400491k>
Koschub T. et al. Biochem Journal 2012: <https://portlandpress.com/biochemj/article-abstract/442/3/483/81274/Allosteric-antibody-inhibition-of-human-hepsin?redirectedFrom=fulltext>
Kromann-Hansen T. et al. Scientific Reports 2017: <https://www.nature.com/articles/s41598-017-03457-7>
Schaefer M. et al. J Mol Biol 2019: <https://www.sciencedirect.com/science/article/pii/S0022283619305625?via%3Dihub>
Ganesan R. et al. Structure 2009: <https://www.sciencedirect.com/science/article/pii/S0969212609004146>

One problem for the case they are presenting is that loop LA doesn't interact with any structural elements of HTRA1 in the apo or active conformations, so, any direct effects on the elements of the activation domain etc are implausible. Note that a plethora of mutations are known to have far-reaching structural effects causing distortions of the active site, which doesn't mean that the underlying mechanism is allostery.

>> We agree that mutations made in the LoopA may not be indicative of an allosteric site and we assured to not raise this perception in the revised Results/Discussion section. The Loop A mutants were made as a result of our structural understanding that the LoopA is the site of origin for the allosteric effect of Fab15H6.v4, rather than to prove that LoopA is an allosteric site. Because the Fab binding to LoopA affects the active site, we reasoned that the LoopA may be a regulator of enzymatic activity. The subsequent experiments with these mutants and with the LoopA swap experiments have confirmed this. We updated the results and discussion sections to reflect this appropriately.

A minor but related issue is that the authors state that the antibody would not cause activation by inhibition. Yet, allostery would predict exactly such an effect. Perhaps I missed to see titrations of the antibody in HTRA1 activity assays including sub stoichiometric concentrations of the antibody.

>> To further address this, we have now performed an HtrA1^{PD} titration assay using sub-stoichiometric concentration of Fab15H6.v4. The results are shown in Supplementary Figure S1e (see below) and demonstrate that HtrA1^{PD} is not activated by the Fab over a wide inhibitor concentration range. The increasing concentrations of Fab resulted in a progressive inhibition of enzyme activity. Complete

inhibition was achieved at a stoichiometry of 1:3 (HtrA1 trimer:Fab15H6.4), consistent with the stoichiometry observed in the structures. We describe this new result in the Result section (page 5) in accordance with the request of another reviewer inquiring about the stoichiometry of the complexes:

“Additionally, titration of HTRA1^{PD} with Fab15H6.v4 showed a progressive inhibition of HTRA1^{PD} enzyme activity at sub-stoichiometric concentration reaching complete inhibition at a 1:3 (HTRA1 trimer:Fab15H6.v4) molar ratio (Supplementary Figure S1e), consistent with the stoichiometry determined by size exclusion chromatography (Figure 1e)”

Supplementary Figure S1e. Titration of 10 nM HTRA1^{PD} trimer with sub-stoichiometric concentrations of Fab15H6.v4 (32 nM to 6.5 nM). Full inhibition is achieved at a concentration of 32.5 nM of Fab15H6.v4, indicating a 1:3 complex (HTRA1 trimer:Fab).

Reviewers' Comments:

Reviewer #1:

Remarks to the Author:

The Authors responded in full and satisfactorily to all my comments.

I have no further critical remarks.

Taking into consideration the responses to the comments of other Referees I think the manuscript has been considerably improved. I see no significant flaws in it.

Reviewer #2:

Remarks to the Author:

The authors appropriately responded to all raised concerns in detail, and the changes and additions lead to an improvement of the manuscript. I have no objections and recommend publication of the revised manuscript without further changes.

Reviewer #3:

Remarks to the Author:

While the response of the authors is appropriate, I'd like to again suggest to be a bit cautious when defining a novel regulatory site as, in my view, an equally likely explanation of the results is that both, high affinity ligands and mutations, could simply interfere with conformational flexibility that is essential for the functioning of the sensitive and complex activation domain of S1 serine proteases during reversible disorder to order transitions. That said, I leave it up to the authors to take the note of caution on board or not, as it is their and not my paper.

REVIEWERS' COMMENTS

Reviewer #1 (Remarks to the Author):

The Authors responded in full and satisfactorily to all my comments.

I have no further critical remarks.

Taking into consideration the responses to the comments of other Referees I think the manuscript has been considerably improved. I see no significant flaws in it.

Reviewer #2 (Remarks to the Author):

The authors appropriately responded to all raised concerns in detail, and the changes and additions lead to an improvement of the manuscript. I have no objections and recommend publication of the revised manuscript without further changes.

Reviewer #3 (Remarks to the Author):

While the response of the authors is appropriate, I'd like to again suggest to be a bit cautious when defining a novel regulatory site as, in my view, an equally likely explanation of the results is that both, high affinity ligands and mutations, could simply interfere with conformational flexibility that is essential for the functioning of the sensitive and complex activation domain of S1 serine proteases during reversible disorder to order transitions. That said, I leave it up to the authors to take the note of caution on board or not, as it is their and not my paper.

We thank the reviewer for this comment, which prompted us to reassess the term “regulation” which we utilize extensively to characterize the role of LoopA for HTRA1 catalytic activity. While the term “regulatory site” expresses the connectivity of LoopA with the active site in a simplified and perhaps somewhat loose manner (in the broadest sense of the word “regulatory”), it also bears the potential of over-interpreting the results since “regulation” in the strict sense defines a physiological cause-and-effect relationship which we have not proved. Therefore, we heeded the words of caution and decided to describe the importance of LoopA for catalytic function in different terms and not suggestive of a regulatory site (using descriptions like “being important/critical/essential for enzyme function” instead).